# Breathable, wearable skin analyzer for reliable long-term monitoring of skin barrier function and individual environmental health impacts

Insic Hong [1,2,6], Daseul Lim [1,6], Dongjin Kim [1,6], Myungrae Hong [1,3,6], Sanghun Kang [1], Kyungbin Ji[1], Taeuk Oh[1], Suhyeon Hwang[1], Yeonwook Roh [1], Dohyeon Gong[1], Gibeom Kwon[1], Taewi Kim [1], Chaewan Im[1], Eunyoung Kim[1], Jingoo Lee[1], Seongyeon Kim [1], Juil Kim[1], Seunghyun Kim[4], Kyungmin Shim[5], Jungho Lee[1], Sungchul Seo [5] ✉, Je-Sung Koh [1] ✉, Seungyong Han [1] ✉ & Daeshik Kang [1,3] ✉

Monitoring skin health through parameters like skin hydration (SH) and transepidermal water loss (TEWL) is vital for diagnosing skin conditions and identifying disease factors. Conventional devices and survey-based methods often fail to deliver accurate diagnoses due to circadian rhythms of skin health data, limited measurement frequency, and patient subjectivity. Previous research has shown that prolonged device usage also causes sweat accumulation, compromising reliable monitoring. Here, we present a breathable skin health analyzer (BSA), a wearable device designed for prolonged use, capable of accurate, long-term measurement of SH and TEWL. The BSA addresses considerable obstacles in skin health monitoring by employing a breathable chamber and a bistable actuator that ensures both ventilation and consistent sensor contact with the skin. Validated through a 28-day clinical trial, the BSA and data processing algorithms demonstrated their effectiveness in providing reliable data by analyzing the correlation between particulate matter exposure and the skin barrier health. These results not only highlight the potential to improve the diagnosis and treatment of diseases but also show the possibility of contributing to individual environmental health impact assessments and translational studies.

The skin, the body's largest organ, interacts with the environment, regulates body temperature and humidity, and protects against pathogens and harmful substances through immune surveillance[1–3]. Because maintaining a healthy skin barrier is crucial, quantitative assessments of skin health—based on key biomarkers such as skin hydration (SH) and transepidermal water loss (TEWL)—are widely used to diagnose skin diseases and identify relevant disease factors[4–8]. However, these skin health parameters exhibit inherent 24-hour

[1]Department of Mechanical Engineering, Ajou University, Suwon-si, Gyeonggi-do, South Korea. [2]Center for Systems Biology, Massachusetts General Hospital, Boston, MA, USA. [3]Department of Mechanical Engineering, Pohang University of Science and Technology (POSTECH), Pohang, South Korea. [4]Department of Medical Humanities, Korea University College of Medicine, Seoul, South Korea. [5]Department of Nano-chemical, Biological and Environmental Engineering, Seokyeong University, Seoul, South Korea. [6]These authors contributed equally: Insic Hong, Daseul Lim, Dongjin Kim, Myungrae Hong. ✉e-mail: haha0694@skuniv.ac.kr; jskoh@ajou.ac.kr; sy84han@ajou.ac.kr; dskang@postech.ac.kr

circadian variability, making short-term measurements often insufficient to provide comprehensive information[9–12]. For instance, SH and TEWL levels can fluctuate throughout the day, with data points potentially reflecting peaks or troughs depending on the time of measurement. This variability is particularly evident in atopic dermatitis patients, where evening exacerbations in TEWL have been observed[13], highlighting the importance of continuous long-term monitoring to accurately assess skin barrier function.

Extensive cross-sectional studies have revealed significant correlations between environmental factors like particulate matter (PM), formaldehyde, and atopic dermatitis (AD)[14,15]. Building on these findings, there is a growing interest in translational research on interventions of environmental diseases, particularly as patients struggle to identify causes of illness individually[16,17]. However, advancing this research faces two key limitations in accurately quantifying skin barrier functions: (1) the absence of long-term monitoring devices capable of capturing daily fluctuations, and (2) reliance on survey-based subjective data. Most commercial instruments are "stick-type", making continuous 24-hour measurements impractical (Supplementary Table 1). Although skin-attachable sensors[18–20] can provide accurate measurements through conformal contact with the skin, sweat accumulation under the device can affect skin physiology[21] during prolonged use.

To address these issues, breathable sensors[22–24] with ultrathin substrates with nano-mesh[25,26] and micro-hole patterns have been developed. While these sensors improve sweat evaporation, they remain vulnerable to mechanical friction and environmental noise, limiting their suitability for everyday use by the general public[23,24,27]. Another approach to prevent sweat accumulation is to design a chamber that allows ventilation at the skin surface[28]. Open-chamber systems allow natural ventilation by exposing the skin to ambient air; however, they are highly susceptible to environmental disturbances such as airflow, which compromises measurement accuracy under daily conditions[29,30]. In contrast, closed-chamber systems offer improved measurement accuracy by isolating the skin from ambient airflows, but they require a mechanism to regularly remove accumulated moisture, leading to increased device bulk and limited applicability for wearable devices[31–33] (Supplementary Table 2). Alternatively, conventional epidemiologic studies have relied on survey-based skin health scoring methods, such as the SCORAD index for atopic dermatitis[34,35]. These methods are limited by subjective factors, including itching and sleep disturbances, which can reduce the consistency and accuracy of the data as they depend on individual emotional responses or experiences[36,37]. To develop a wearable device capable of long-term monitoring, it is essential to miniaturize the chamber and actuator to prevent sweat accumulation, enable monitoring through wireless communication, ensure low power consumption for extended use, and incorporate the ability to measure objective indicators of the skin.

In this work, we developed a breathable skin health analyzer (BSA) that not only reliably measures SH and TEWL but also offers exceptional durability for long-term, continuous monitoring. For clinical validation, we performed continuous monitoring over 28 days to ensure comfort and reliability for everyday use (Fig. 1a). When the skin barrier is damaged, as commonly observed in AD, SH decreases and TEWL increases, indicating a disruption in the lipid matrix of the stratum corneum (Fig. 1a). By simultaneously measuring two inversely correlated parameters in a damaged skin barrier, the BSA enables more accurate diagnosis of barrier function, avoiding reliance on single-parameter assessments.

The BSA comprises a sensor module, control board, actuator, breathable chamber, all compactly and comfortably worn on the body using a low-irritation strap band compatible with commercial (Fig. 1b and c). For easy maintenance, the board and sensor module are connected via a standard board-to-board connector, and the sensor module, comprising a chamber and sensor FPCB integrated within the protective cap, is physically attached to and aligned with the actuator using two pairs of magnets, allowing rapid replacement without damage. For continuous and accurate measurement and ventilation cycles, the sensor module and the surrounding chamber require a compact actuator capable of repeatedly attaching them to and detaching them from the skin. To ensure compactness and energy efficiency for long-term use, we implemented a shape memory alloy (SMA) wire with a bistable beam. SMA, with its exceptionally high power density of up to 5 kW/kg, is a promising candidate for integration into a compact wearable device. During measurement, the actuator presses the sensor against the skin with consistent force, minimizing measurement errors. The actuator also closes the breathable chamber to block external interference during measurement. After measurement, it retracts the sensor and the chamber, allowing moisture at the skin-sensor interface to evaporate naturally and preventing sweat accumulation (Fig. 1d). As shown in Fig. 1e, SH is measured based on transient thermal transfer, enabling more accurate measurements compared to the conventional electrical methods, which are prone to interference from cosmetics, sweat, and environmental noise (Supplementary Table 3). TEWL is measured by a humidity sensor inside a breathable chamber with holes on the sides, which reduce environmental noise, resulting in more robust measurements compared to open-chamber methods (Supplementary Table 4).

By leveraging long-term data collected from the BSA, we aim to identify correlations between skin health and individual-level environmental pollutants and provide evidence-based feedback to patients. To facilitate long-term monitoring and data collection, the BSA integrates Bluetooth communication, battery charging, and sensor/actuator control within a compact form factor (Fig. 1g), weighing only 13 g and sized similarly to a smartwatch—well-suited for routine daily use. Detailed size information about the BSA can be found in Supplementary Fig. 1. Measurement data are stored on the smartphone and transmitted to the server for outlier removal and correlation analysis (Supplementary Fig. 2). Compared to previously reported breathable skin health monitoring sensors[24,25], our BSA provides superior environmental noise isolation, long-term monitoring capability, breathability, wireless operation, and multi-parameter sensing. This long-term skin health monitoring system continuously tracks changes in skin condition, providing more accurate skin health data and enabling earlier detection and management of environmental diseases. Ultimately, the BSA contributes to improving prevention and treatment strategies in managing diseases related to environmental pollution.

## Results

To reliably monitor skin health, it is essential to minimize interference from external factors like temperature, airflow, and humidity, while also preventing moisture accumulation that can lead to TEWL rebound[25,38]. The BSA addresses these needs by integrating SH and TEWL sensors inside a specially designed elastomer-based breathable chamber. The breathable chamber remains open to the environment and closed during measurements to minimize external interference (Fig. 2a). On the top of the chamber, a cap that secures a flexible printed circuit board (FPCB) is attached. The sensor part on the FPCB comprises the SH sensor that utilizes thermal characteristics according to the SH level, and the TEWL sensor that measures water loss through the skin's barrier layer, as shown in Fig. 2b (details of the BSA sensor fabrication procedures in Supplementary Fig. 3 and Methods).

The SH sensor includes two pairs of NTC temperature sensors and a pair of heaters. One pair of NTCs (NTC 2) is placed above the heaters, while the other (NTC 1) is located 1.5 mm away to form a temperature gradient reference (Supplementary Fig. 4). When the SH sensor contacts the skin, the SH sensor applies heat to the resistance heating element (heater) and measures the temperature ($T_1$:NTC 1, $T_2$: NTC 2) at NTCs.

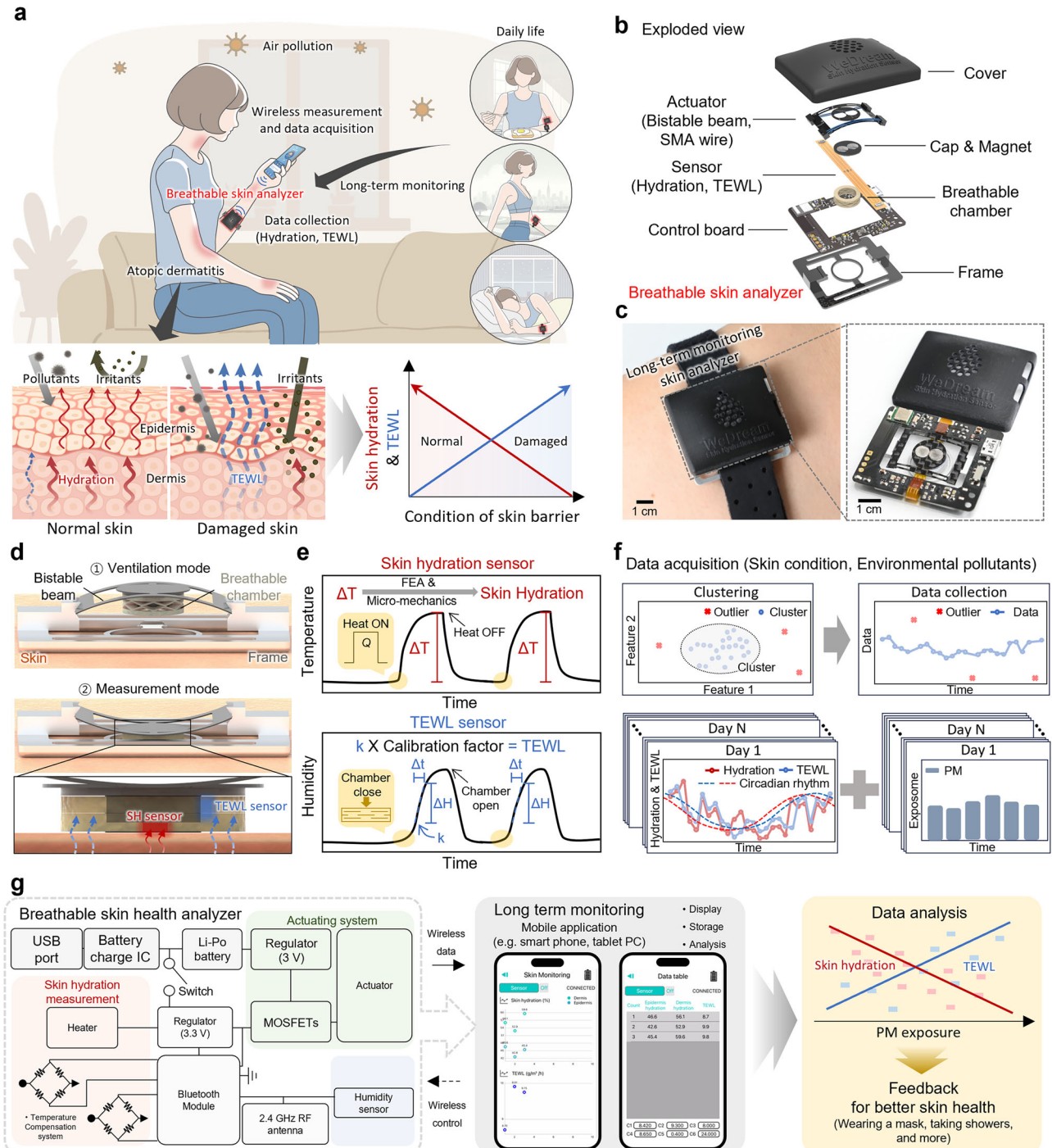

**Fig. 1 | Overview of the breathable skin health analyzer. a** Illustration of SH and TEWL measurement using the breathable skin health analyzer in daily life. **b** Exploded-view schematic of the breathable skin health analyzer with three functional parts: (1) actuator which consists of bistable beam and SMA wire, (2) sensor part with the SH sensor and TEWL sensor, and (3) control board. **c** The assembly of the analyzer and strap band as a forearm wrap. **d** Schematic illustration of ventilation and measurement modes of the actuator for acquiring reliable data. **e** Illustration of measurement principles for SH and TEWL. **f** Diagram indicating data analysis for outlier removal and circadian rhythm for the correlation analysis on skin health diagnosis and individual-level environmental pollutant effects. **g** Electrical architecture of the device, including power management circuits, sensors, and actuator control system.

Details of the sensor components are provided in the top section of Supplementary Table 5. $\Delta T_{12}$, defined as the differential temperature between $T_1$ and $T_2$, is relatively insensitive to external factors such as ambient or substrate temperature, and predominantly reflects local changes associated with evaporation rates due to skin hydration levels. The temperature-compensation strategy enhances the robustness of the sensing system against environmental disturbances and skin temperature, as validated through additional experiments presented in Supplementary Fig. 5. Using transient heat transfer, we derive the thermal characteristics of the skin, such as thermal conductivity (k) and thermal diffusivity (α), and from these, the skin's moisture content ($\phi_s$). The thermal response of the skin was measured using Wheatstone bridges and NTC thermistors. Based on the measured thermal response, the thermal properties of the skin were calculated through finite

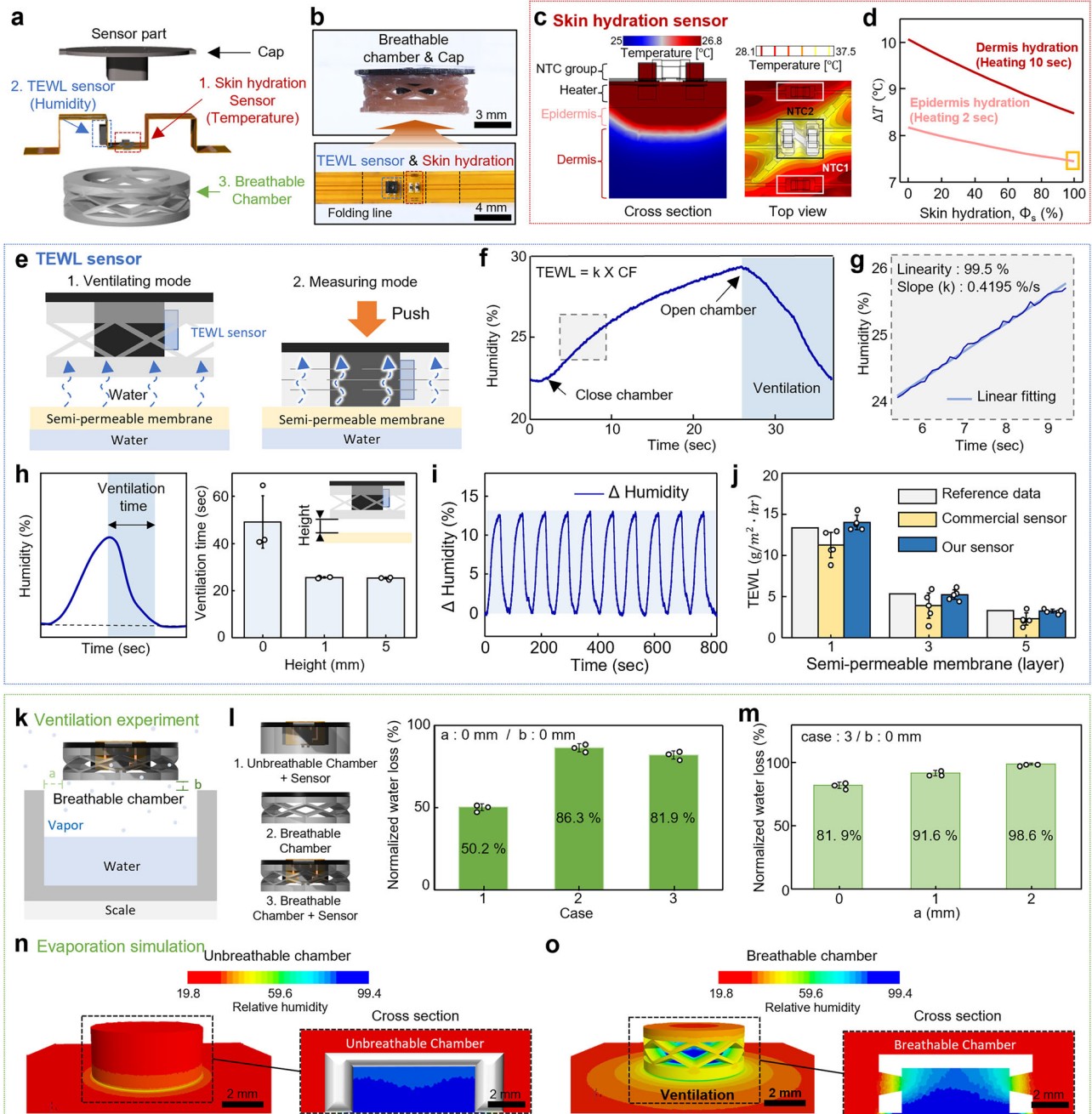

**Fig. 2 | Design of the sensor part in the device for reliable SH and TEWL measurement. a** Exploded-view illustration of the sensor part: a cap, SH sensor, TEWL sensor, and a breathable chamber. **b** An optical image of the sensor part. **c** Cross-sectional and top view of the temperature distribution induced by the heaters on the skin for 2 s. **d** Transient temperature ($\Delta T$) response generated by different heating time on SH. **e** Illustration of the wet-cup method set up to generate different water vapor flux values for calibration of the TEWL sensor. **f** Graph of humidity change inside the chamber due to the closed breathable chamber. **g** Calculating of TEWL using the linear squares method. **h** Graph of ventilation time according to chamber height from semi-permeable membrane ($n = 3$). $n$ represents the number of technical independent replicates. Data were expressed as mean ± SD. **i** Repeatability of measurements of the TEWL sensor. **j** TEWL measurement using reference data (wet-cup method), a commercial sensor and our sensor according to the number of semipermeable membrane layers ($n = 5$). $n$ represents the number of technical independent replicates. Data were expressed as mean ± SD. **k** Illustration of the ventilation experiment. Change in the amount of water evaporation depending on the vertical (**l**) and horizontal distance (**m**) ($n = 5$). $n$ represents the number of technical independent replicates. Data were expressed as mean ± SD. Evaporation simulation results for the unbreathable chamber (**n**) and breathable chamber (**o**).

element analysis and the Maxwell-Eucken model, enabling the estimation of skin hydration levels. Detailed information of processes to calculate moisture content ($\phi_s$) is provided in Supplementary Note 1. Fig. 2c shows a finite element analysis (FEA) demonstrating how the temperature distribution changes with SH level upon heating (Supplementary Fig. 6 and Supplementary Table 6). By adjusting heating time— 2 seconds for epidermal hydration ($\phi_E$) at ~100 μm depth and

10 seconds for dermal hydration ($\phi_D$) at ~1400 μm depth[19]—we can analyze both the epidermal layer and the dermal layer to provide a more accurate assessment of skin hydration. Figure 2d presents that the epidermis and dermis allow different $\Delta T$ depending on the SH level ranging from 0 to 100 %. The results can derive $\Phi_E$ and $\Phi_D$ by applying the micromechanical modeling that relates to the temperature sensor to SH. (Supplementary Fig. 7).

TEWL is a key indicator for evaluating the integrity of the skin barrier. In patients with atopic dermatitis, a compromised skin barrier results in increased trans-epidermal water loss and increased permeability to external irritants, potentially aggravating symptoms. The TEWL measurement of the BSA uses a closed chamber method (CCM), which measures changes in moisture evaporation from the skin within a sealed chamber over a specified period, using a humidity sensor. Unlike the open-chamber approach, CCM provides stable and reliable measurement less influenced by environmental changes[39]. During measurement, the breathable is closed under vertical force, and the humidity sensor inside the chamber measures the rate of moisture increase. After measurement, the chamber reopens, allowing ventilation and ensuring that repeated measurements remain accurate and unaffected by residual moisture (Fig. 2e).

We validated the TEWL sensor by analyzing humidity changes across a semi-permeable membrane (Fig. 2f). When the chamber is closed by a vertical load, the relative humidity increases to 29 %. After 25 seconds of being closed and upon removal of the external force, the chamber opens and the humidity drops back to the initial humidity value of 23 % through ventilation via the hole of the chamber. TEWL is calculated by multiplying the slope of increasing humidity ($k$), determined to be 0.42%/s within the 6 to 9 second interval, where linearity reaches 99.5% as assessed using the least squares (LS) method, with a calibration factor (CF) as shown in [40] Fig. 2g. To determine CF, we applied the wet-cup method to measure the $k$ value, which was matched to reference TEWL values obtained using water at 23 °C and 40 °C ($5.1 \pm 0.4$ and $23.7 \pm 0.9$ g/m²·h, respectively). Detailed procedures are provided in Supplementary Note 2. The corresponding $k$ values were 0.21 and 0.98, from which the calibration factor was calculated using TEWL = $k \times$ CF, yielding an approximate value of 24. Figure 2h shows how the ventilation time changes depending on the distance from the semi-permeable membrane and shows the minimum distance required for proper ventilation. The ventilation time is defined as the duration for the humidity level to the initial humidity value after chamber opening. When the breathable chamber is placed directly on the semi-permeable membrane (0 mm distance), twice as much ventilation time (50 seconds) is required compared to when the chamber is placed 1 mm away from the membrane. The ventilation time for the distances of 1 mm and 5 mm was lower than 30 seconds with negligible difference and error ($n = 3$). The TEWL sensor and the breathable chamber placed at the minimum distance (1 mm) from the semi-permeable membrane show reliable repetitive TEWL measurement (Fig. 2i). The result shows that the humidity recovered to the initial value for every cycle after 30 seconds of ventilation time without the accumulation of moisture. In addition to one layer of the semi-permeable membrane, we measured the humidity with 3 and 5 layers of membranes to verify sensing performance compared to the reference data and commercial sensor (GP skin) as shown in Fig. 2j ($n = 5$). As the number of layers increased, reference data and commercial sensors showed reduction in TEWL. Reliable ventilation and repeated measurement cycles confirmed that the BSA's TEWL sensor closely matches reference data and a commercial sensor, demonstrating consistent, high-quality performance.

For reliable long-term monitoring of atopic dermatitis, ensuring high ventilation at measurement sites is critical to prevent skin irritation[41]. We conducted design analysis of the breathable chamber to enhance the breathability of the measurement area (Fig. 2k–o) and confirmed that introducing ventilation holes in the chamber significantly improved breathability. To assess the impact of the sensor parts including the chamber on the skin water evaporation, the continuous water loss from a 10 ml bottle was measured using an ultra-precision scale across various sensors configurations (Fig. 2k). Normalized water loss was measured compared to the pristine state on the water bottle, breathable chamber and a sensor integrated chamber (Fig. 2l). The chamber without holes on the side was 50.2% and the breathable chamber was 81.9%, which was an improvement of about 1.6 times in breathability. Additionally, an experiment was conducted to enhance ventilation by increasing the horizon distance (a) in breathable chamber (case 3) (Fig. 2m). The result reveals that by achieving 98.6% evaporation, which is equivalent to that in the pristine state when placed 2 mm away. Accordingly, the BSA was designed to float more than 2 mm from the measurement site for maintaining high ventilation. Figure 2n–o shows the demonstration of the breathability effect of the chamber using the natural evaporation simulation of STAR-CCM+. A general chamber maintains a constant relative humidity inside. In contrast, the breathable chamber represents a gradient of relative humidity inside, which occurs through a hole leading to the outside atmosphere by ventilation. The results indicate that the sensor part of the BSA was designed to ensure reliable measurement by verifying the performance through simulation and experiment.

We analyzed simulations of the chamber on the skin and the actuator parameters to ensure that the breathable chamber reliably opens and closes on deformable skin. Considering the skin's stiffness, both the chamber's stiffness and that of the actuator must be optimized to minimize skin deformation and reduce measurement errors. The breathable chamber is opened and closed on flexible skin using shape memory alloy (SMA)-based bistable actuators. The combination of SMAs and a bistable structure is well-suited for applications requiring a light, compact form factor and with high energy efficiency and high-power actuation[42,43]. We verified the force required to close the chamber and assessed its behavior on the flexible substrate (muscles, fat, skin) through simulations and experiments (Fig. 3a). Simulation results showed that a force of approximately 0.17 N caused the chamber to deform 1 mm in height and fully close (Supplementary Fig. 8). This trend matched that observed in actual samples, with a constant stiffness of 0.167 N/mm until the chamber fully closes. After complete closure, the stiffness increases to 1.765 N/mm. The volume of the chamber decreased linearly during closing and the rate of volume reduction substantially slowed down after closure. (Fig. 3b). Ideally, skin deformation should remain minimal before the chamber closes. If the chamber's stiffness is too high, pressing it against the skin could lead to excessive deformation, thus requiring different actuating stroke depending on user skin. Considering the skin's stiffness (1.6-2.7 N/mm), the pre-closure stiffness of 0.167 N/mm allows for negligible skin deformation until the chamber fully closes. Detailed information of the process to calculate stiffness and deformation of skin is provided in Supplementary Note 3 and Supplementary Fig. 9.

The actuation system consists of a compliant beam (CFRP), SMA and an aluminum frame to open and close the chamber (Fig. 3c). The design incorporates two SMA wires and a CFRP compliant beam attached to an aluminum frame, wherein the beam is bent (details of the actuator fabrication procedures in Supplementary Fig. 10 and Methods). SMA wires are placed either on the upper or lower side of the compliant beam with the interlocking joints (Fig. 3d and e). Each actuator triggers snap-through bistable transition between the beam's downward and upward stable states (Fig. 3f).

Figure 3g illustrates the SMA-based bistable actuator's working principle. In its initial state (state 1), the compliant beam, longer than $L_{frame}$ and bent upwards, is assembled (Supplementary Fig. 11). At this stage, no pressure is applied to the chamber, allowing it to remain open above the skin for ventilation (Supplementary Movie 1). The bent beam applies tension to the attached SMA wire 1. When applied by current, SMA wire 1 contracts and the contraction forces the compliant beam to bend in the opposite direction (state 1 → 2). If the SMA wire contracts sufficiently to exceed a critical point, the beam's bistable nature causes it to snap through and bend in the opposite direction (state 2), closing the chamber under pressure. This pressure ensures that sensors within the chamber can measure skin temperature and humidity accurately, minimizing external influences and enhancing sensor precision. When current is applied to SMA wire 2, it contracts, returning the beam to its original state (state 2 → 1) and allowing

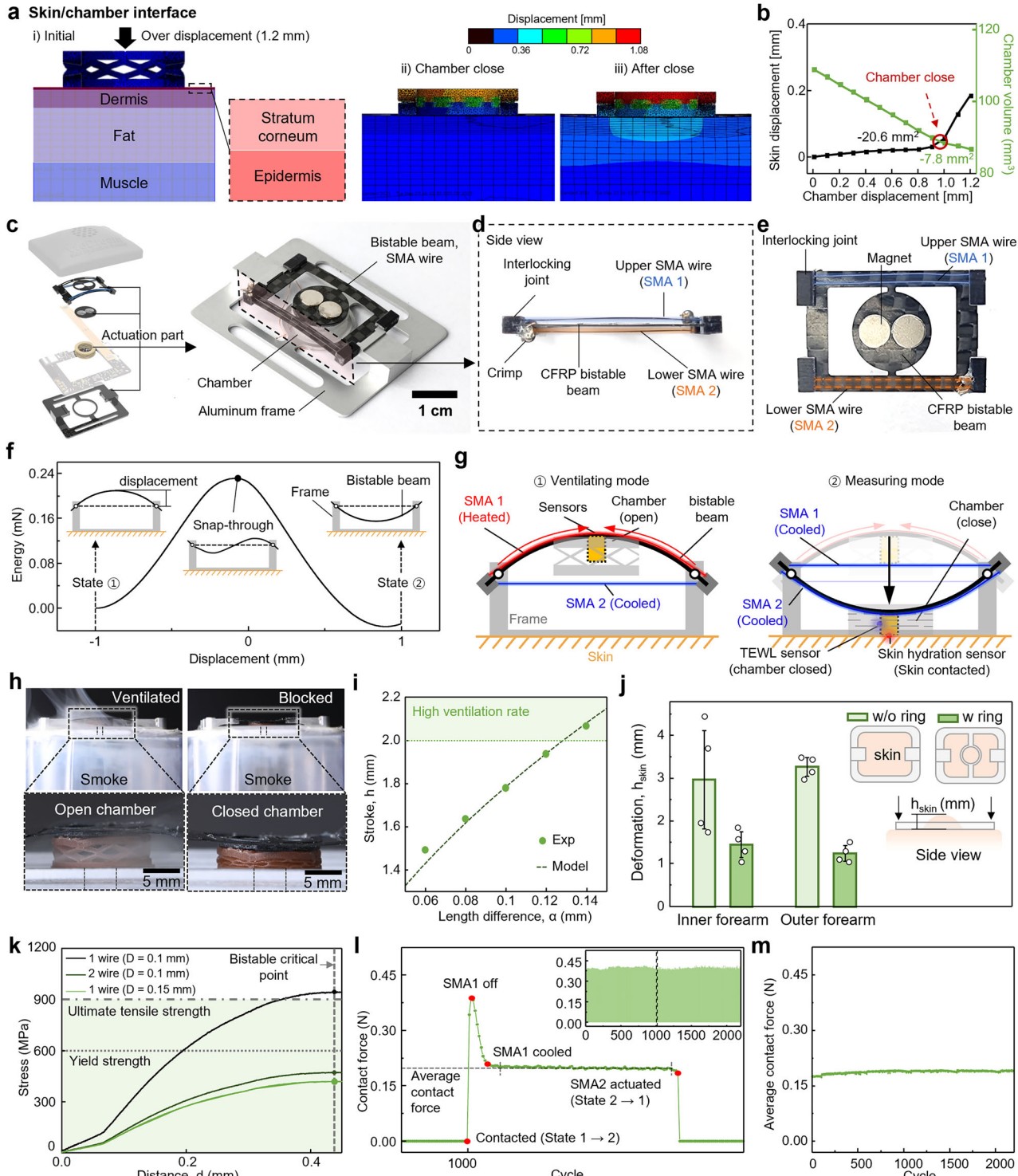

**Fig. 3 | Mechanical behavior analysis between the breathable chamber and the skin, and the corresponding actuator design. a** FEA analysis results of the stress distribution in the breathable chamber under a vertical load. **b** Graph of skin displacement and chamber deformation according to chamber displacement. **c** Image of the actuator combining a bistable beam, SMA wire, chamber, and frame. **d** Side view, and **e** top view of the actuator in its initial state before being fit into the frame. Upper and lower SMA wires are mounted on both sides of the beam. **f** Energy-displacement curve of the bistable beam, showing the two stable minimum energy state (① and ②) and an intermediate unstable state (snap-through). **g** Schematic of the side view of the actuating part demonstrating the actuation and sensing mechanism. **h** Images showing the difference in smoke ventilation when the breathable chamber is open versus closed. **i** Actuation stroke of the bistable beam for the length difference between the frame and the bistable beam. **j** The degree of skin deformation with and without the ring in the frame. **k** Stress profile as a function of distance based on the characteristics of the SMA wire ($n = 4$). $n$ represents the number of technical independent replicates. Data were expressed as mean ± SD. **l** Changes in contact force during cyclic testing of the bistable actuator and SMA operation. **m** Average contact force of the bistable beam over cycles.

ventilation between the sensor and the skin. The bistable actuator, which only requires input energy to shift its state and no energy to maintain each state, is suitable for energy-efficient long-term use. Figure 3h visually demonstrates the chamber's ventilation and sealing capabilities, showing smoke escaping when open and staying contained when closed.

Optimizing the actuator's design parameters can minimize the necessary actuator size. A stroke of approximately 2 mm is required to close the chamber for proper ventilation (Fig. 2m). This is determined by the difference ($\alpha$) between $L_{beam}$ and $L_{frame}$ lengths. Ideally, when $L_{beam}$ and $L_{frame}$ lengths are equal, the beam does not bend, and the stroke is 0. As $L_{beam}$ length increases, the beam bends more (Fig. 3i). The model allows for the determination of $L_{beam}$ (Constant $L_{frame}$) required for the desired actuator stroke. With $L_{frame}$ at 9.47 mm, $L_{beam}$ needs to be 0.14 mm longer for a 2 mm actuator stroke. For stable opening and closing of the chamber, it is crucial to maintain a constant distance between the actuator and the skin. When the device is strapped to the skin, pressure causes the skin to rise into the aluminum frame's empty space. The height of this rise ($h_{skin}$) can be affected by the area of the frame's empty space where the skin deforms upward. Adding a ring-shaped structure around the measurement area on the aluminum frame limits skin deformation, reducing it by about twice compared to when the ring is not present (Fig. 3j).

To enhance the bistable structure's operational cycle, the actuator design must ensure that the maximum stress required for the bistable beam to change states does not exceed the material strength of the SMA actuator. Reducing stress can be achieved by increasing the wire's diameter or the number of wires used. As seen in Fig. 3k, stress of a single wire with a diameter of 0.1 mm exceeds the SMA's ultimate tensile strength. In contrast, using two wires or a diameter of 0.15 mm results in stress below the yield strength, ensuring safe actuator operation. The actuator was designed to operate safely for over 2000 cycles by using two wires with a diameter of 0.1 mm (Fig. 3l). The actuator also allows for consistent pressure application on the skin, maintaining an average pressure of about 0.18 N across more than 2000 cycles (Fig. 3m). We also verified that the stress and strain in the sensor components remain within the material limits under the maximum load applied by the bistable beam (Supplementary Fig. 12).

Figure 4 shows the wireless system and demonstrates the verification of the BSA performance through an in vivo test. The BSA consists of a control board (details of the board design in Supplementary Fig. 13 and the bill of materials for all electronic components is provided in the bottom section of Supplementary Table 5) that includes a Bluetooth chip with an integrated SH sensor, TEWL sensor, a cover that protects the FPCB for everyday use and actuator, powered by a 250 mAh lithium-ion battery (maximum operating time of 36 hours), as shown in Fig. 4a. The robust design and integrated system minimize user fatigue during long-term monitoring, allowing convenient measurements without complex protocols or specialized training. Figure 4b shows the operation scenario of the BSA to measure SH and TEWL on the skin. When the BSA is activated and connected to the mobile phone app via Bluetooth, an actuating voltage of 3 V is applied after 5 seconds delay to stabilize the system (Supplementary Movie 2). The contraction of the SMA 1 results in the downward actuating of the bistable beam, which closes the breathable chamber. TEWL is then measured over a 10 second period. Subsequently, the heater is activated for an additional 10 seconds to measure SH. After completing these measurements, the voltage is applied to trigger SMA 2, which switch the bistable structure to bend upward. The actuation opens the chamber, facilitating air circulation within and ventilation at the measurement site for a duration of 5 minutes. This cycle repeats continuously. Switching between ventilation and measurement modes using the bistable actuator and breathable chamber reduces user variation and external interference (Fig. 4c). The BSA showed significantly lower error (~3.4%) compared to commercial sensor 1

($\phi_{commercial\ 1}$, Corneometer) and 2 ($\phi_{commercial\ 2}$, GP skin) when SH measuring various sites of the body 10 times repeatedly (Supplementary Fig. 14 and Supplementary Fig. 15).

We examined the initial change in SH when subjects wore the non-breathable patch type SH sensor developed in previous research, and the BSA (Fig. 4d and Supplementary Fig. 16). While the patch type SH sensor without the ventilation system increases by up to 15% from the initial SH due to poor ventilation and sweating accumulation, the BSA is positioned 1 mm from the skin, featuring a breathable chamber that facilitates maintaining initial humidity. It demonstrates 98% agreement with the initial SH. Similar to SH sensors, the high ventilation and breathability structure of the BSA proved negligible effects on the skin surface micro-climate during TEWL measurements (Fig. 4e). Repeated measurements maintained initial humidity levels with only about a 3% change. Additionally, we conducted tape stripping experiments to ensure reliable measurements of TEWL. The sensor presents quantified increases in TEWL as stratum corneum layers are progressively removed via the tape stripping (Supplementary Fig. 17). Because the BSA applies consistent force, it accurately measures TEWL with the closed chamber in contact with the skin. The commercial sensor exhibited a high error (~25%) when three experimenters repeatedly measured the forearm of a subject with healthy skin, due to variations in the contact angle and applied pressure of the sensor for each person (Supplementary Fig. 18). Conversely, the BSA enables the measurement of TEWL with high accuracy (standard deviation: ~ 4 %).

Figure 4f shows the results of SH and TEWL measurements throughout the day using BSA and a comparison to a commercial TEWL sensor. The BSA consistently provides reliable data, enabling continuous monitoring of skin health. During monitoring, both BSA and commercial devices show outliers with high TEWL values after a shower or when walking. As reported clinically, the accuracy of measurements cannot be assured when skin conditions are substantially different from daily routine or when ambient humidity or temperature changes rapidly[44,45]. Therefore, we developed an algorithm to remove these outliers. Figure 4g shows conditions where TEWL cannot be accurately measured because the skin is abnormally moist from showering, sweating, or rapidly changing indoor/outdoor humidity. To remove outliers, DBSCAN, one of the density-based clustering algorithms, was used. We collected a total of 98 normal and abnormal data by moving indoors and outdoors and artificially spraying water on the skin. Also, the sensor operation scenario was modified to collect initial humidity data prior to the TEWL measurement. The collected initial humidity and its difference from the previous initial humidity can be clustered using the DBSCAN algorithm as variables. Figure 4h shows the data when the microclimate changed during the measurement, causing an outlier and the normal data measured under routine activities. Figure 4i is the result of clustering, where regions with a high difference from the previous initial humidity, regardless of initial humidity, are clustered −1, regions with low initial humidity and a low difference from the previous initial humidity are clustered 0, and regions with high initial humidity and a high difference from the previous initial humidity are clustered 1. The two graphs show that the clustering algorithm is working well, with only six data points that are incorrect clustering (Failed to cluster ground truth outliers as outliers and ground truth normal data as normal data). The rand measure index, Jaccard index, and Fowlkes-Mallows index values for this algorithm are 0.94, 0.93, and 0.96, respectively, indicating that the similarity between the clustering algorithm and the actual measured data is higher than 0.93. (Supplementary Tables 7 and 8) Therefore, the DBSCAN algorithm can be used to remove outliers to provide users with refined data.

Current skin health diagnostic devices often face issues with diagnostic accuracy due to limited measurement frequency and reliance on a single measurement parameter. The limitations make it difficult to distinguish between atopic dermatitis patients and healthy individuals based on TEWL and SH values because there are

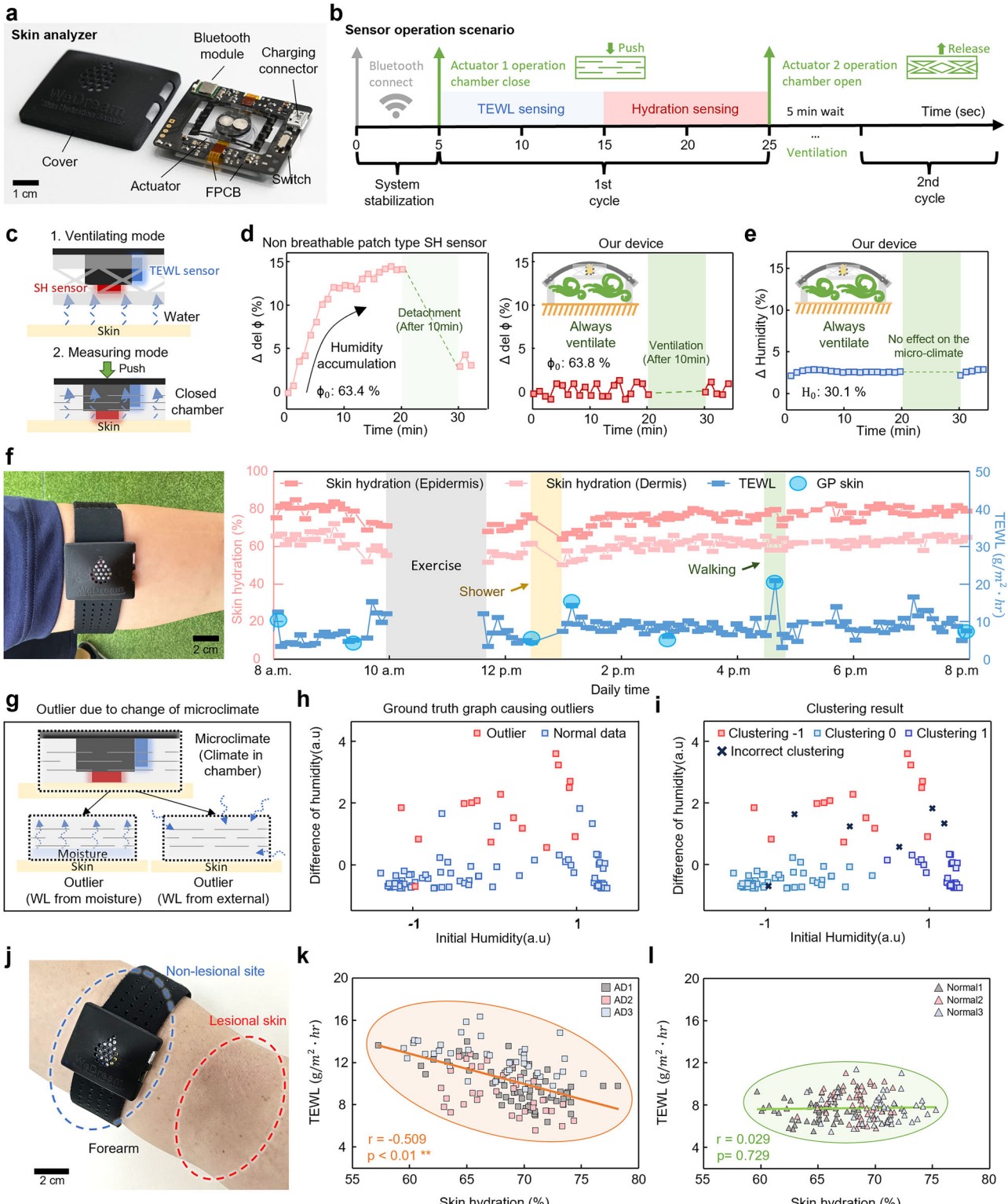

**Fig. 4 | Demonstration of the BSA. a** An optical image of the unencapsulated device. **b** Diagram of the sensor operation scenario. **c** Illustration of the ventilation and measurement mode of breathable chamber by the actuator. **d** Comparison of the SH increase in long-term monitoring between non breathable patch-type SH sensor and our device. **e** Graph of the impact of micro-climate on TEWL measurement in our device during long-term monitoring **f** Measurement of SH and TEWL using the BSA during a day in daily life. **g** Illustration of when clinically outliers occur. **h** Ground truth graph of outlier and data when using humidity and humidity difference as features **i** Clustering graph of outlier and data when using humidity and humidity difference as features **j** An optical image of the device worn on non-lesional site near the lesional skin. **k**, **l** Comparison of the correlation between SH and TEWL for both atopic dermatitis patients and normal subjects. Statistical significance was assessed using two-sided Pearson's correlation analysis.

overlapping measurement ranges of TEWL and SH between the two groups[46–48]. By measuring SH and TEWL over an extended period and using their inverse relationship, the BSA can more accurately assess skin barrier damage.

As a demonstration, BSA was clinically tested on three patients with mild to moderate atopic dermatitis and three healthy people, monitored for three days. It was worn on a non-lesional site near the lesional skin to prevent skin conditions from worsening (Fig. 4j). This clinical protocol was guided by previous studies showing that SH and TEWL in normal skin near the lesion exhibit consistent trends[49–51]. The measurement data collected over three days were processed as hourly means and subsequently subjected to bivariate correlation analysis. In the AD group, the statistical analysis revealed a negative correlation between these measures, with a Pearson correlation coefficient $r = -0.509$, indicating a moderate inverse relationship (Fig. 4k). This result is statistically significant with $p < 0.01$, consistent with established findings that damage to the skin barrier results in increased TEWL and decreased hydration[44,48]. In contrast, no statistically significant correlation was observed between the factors in the control group (Fig. 4l). Statistical comparisons between non-lesional skin of AD patients and healthy skin, along with the correlation analysis for each individual in the clinical and control groups is, are detailed in Supplementary Fig. 19. These results emphasize BSA's unique ability to assess the health of skin barrier with statistical significance at the individual level.

As a second demonstration, we propose that the BSA's ability to provide long-term, reliable measurements is essential for assessing environmental health impacts. Environmental diseases are health disorders linked to physical, chemical, and biological factors in the environment. Recent studies have discovered that air pollution, particularly PM, can penetrate the skin barrier or respiratory system, triggering cytokine responses and intensifying AD[52,53]. Given that, our BSA provides long-term measurement techniques that can quantitatively analyze the correlation between environmental factors and AD.

Through physician interviews, we found that patient 2 of three clinical patients had worsening AD symptoms related to PM exposure. This observation led us to initiate a longitudinal study to systematically examine the effects of PM on skin barrier health. The clinical trial was conducted in April, when PM concentration in South Korea is at its highest due to the yellow dust phenomenon. A subject wore a portable PM sensor and BSA for 14 days to monitor the condition of the skin barrier in response to PM exposure. (Fig. 5a and Supplementary Fig. 20). To accurately assess the impact of PM exposure on skin health, it was crucial to consider the inherent circadian variations in biological processes influencing skin condition. These natural fluctuations could mask the true effects of environmental exposures if not properly accounted for. Therefore, data processing involved analyzing daily variations through circadian rhythm analysis to derive the MESOR (midline estimating statistic of rhythm) and amplitude representing daily values of TEWL and SH (Fig. 5b). For PM exposure, we calculated the total exposure over each day (Supplementary Fig. 21). The BSA measured SH and TEWL, and the portable PM sensor measured the concentration of PM, ambient temperature, and relative humidity. Additionally, the SCORAD index, widely used in clinical trials, is recorded daily to document the severity of skin barrier damage.

Figure 5c displays a heat map showing the correlations among various factors measured over 14 days, with statistically significant correlations marked with an asterisk where the $p$-value is less than 0.05. It shows a negative correlation between MESOR of hydration (MH) and MESOR of TEWL (MT) due to skin barrier damage, and a statistically significant increase in MT with increased PM exposure (PME). The SCORAD index (SI) shows statistical significance with increased mean relative humidity (RHM), suggesting a potential bias in the perception of greater skin damage with increasing humidity. Key factors graphed include the relationship between TEWL and SH, which reflects the presence of skin barrier damage, as well as the relationship

between exposure to particulate matter and both SH and TEWL (Fig. 5d–f). While the TEWL measurements taken by the BSA demonstrated statistically significant correlations with PM exposure, the SCORAD index did not show statistical relevance, possibly due to its dependence on subjective assessments such as symptom evaluation like itchiness and sleep disturbances (Supplementary Fig. 22a). This suggests that variables like SCORAD, which incorporate subjective components, require larger sample sizes to reliably evaluate the extent of skin damage caused by PM exposure[54–56]. While this makes SCORAD well-suited for large-scale epidemiological studies, it may be less applicable for analyses based on smaller datasets at an individual level. Figure 5g is a graph that analyzes the lag time correlation of SH and TEWL when exposed to PM. As for the data processing method, data from 14 days of measurements were averaged over 3-hour periods to analyze the timing of PM effects, and the data were categorized into PM exposure levels of 0 to 40 mg/m³·day and 40 to 80 mg/m³·day to analyze the impact of PM concentration. The Spearman correlation coefficient, a rank-based metric, was used to indicate how the correlation between data changes over time. In the PM exposure range of 0 to 40 mg/m³·day, the spearman correlation coefficient values between SH are −0.25, −0.32, and −0.24 for 0–3, 3–6, and 6–9 hours, respectively, and between TEWL are −0.16, −0.13, and −0.12 for 0–3, 3–6, and 6–9 hours, respectively (Supplementary Fig. 23). Consequently, when PM exposure is 0 to 40 mg/m³·day, no statistically significant lag time correlation is observed between PM exposure and changes in SH or TEWL. On the other hand, when PM exposure is relatively high, at 40 to 80 mg/m³·day, an increase in the correlation coefficients was observed. This suggests that higher PM exposure levels result in more immediate changes in SH and TEWL. These findings are consistent with results from previous studies where direct exposure to harmful agents on the skin induced similar effects, suggesting the possibility of a threshold at which SH and TEWL respond to PM concentration.

Based on clinical results and the correlation between PM exposure and skin barrier damage, interventions were implemented. From various intervention strategies suggested in previous research, we adopted suitable methods for individuals, such as wearing masks while outdoors and showering upon returning home (Fig. 5h). The bottom graph in Fig. 5h shows that PM concentrations remained constant throughout the entire clinical period. This consistency in PM exposure ensures a reduction in potential confounding variables. Graphs showing changes in SH, TEWL, and PM matter exposure over 14 days post-intervention, along with a heat map displaying correlations among eight factors, are detailed in Supplementary Fig. 24. Analyzing the correlations among key factors reveals that the MESOR of SH and TEWL, which indicate skin damage, consistently show a negative correlation, suggesting persistent damage to the skin barrier (Fig. 5i). However, the MESOR of TEWL in response to particulate matter exposure have shifted to a weak negative correlation with no statistical significance. Additionally, the relationship between SH and SCORAD indices showed a different trend after the intervention with respect to PM exposure, but it was not statistically significant (Fig. 5j, k and Supplementary Fig. 22b). Furthermore, as shown in Fig. 5l, it was observed that there was no statistically significant time lag correlation in changes in SH or TEWL in either the 0 to 40 mg/m³·day or 40 to 80 mg/m³·day PM exposure ranges. These findings suggest that the intervention successfully mitigated the deterioration of the skin barrier due to PM, and highlight the effectiveness of the BSA in managing environmental health at an individual level.

## Discussion

Current skin health measurement devices are limited to short-term monitoring or designed to function only in specific environments with minimal external noise, posing constraints on practical, stable long-term skin health monitoring. Most commercial stick-type devices are prone to measurement errors due to manual handling and are impractical for

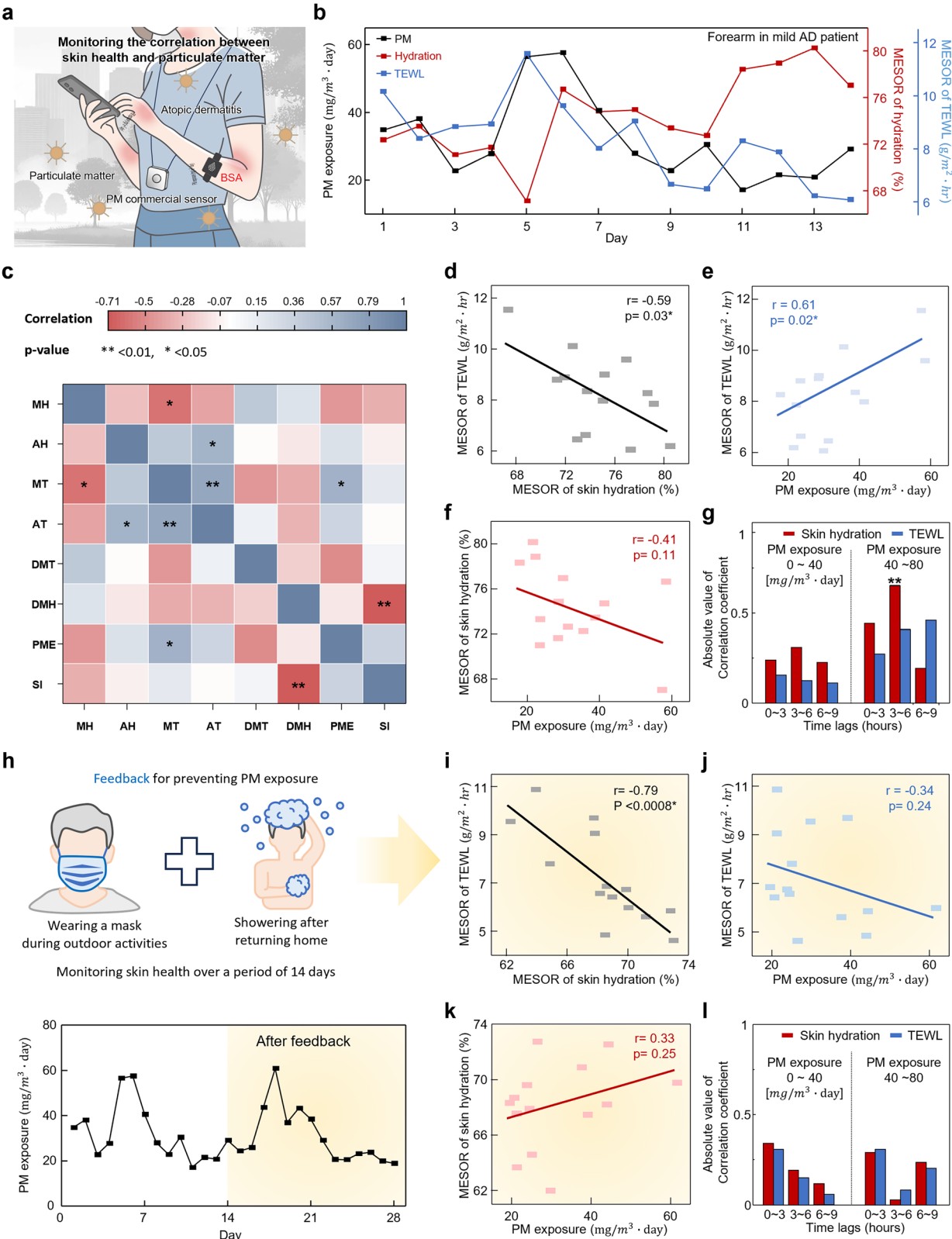

continuous daily use during routine activities. Additionally, thin and flexible wearable sensors that conform to the skin are highly susceptible to measurement errors from external noise, lack proper ventilation for sweat, and are vulnerable to friction from outdoor activities, making them suitable only for short-term and controlled environments.

Our device addresses these limitations by incorporating a breathable chamber that opens and closes using a thin-form-factor

SMA-based bistable beam actuator, which blocks external noise during measurement and provides ventilation when needed. This ventilation prevents moisture accumulation from sweat on the measurement area, enhancing measurement accuracy and extending the device's usable duration. The actuator also applies consistent pressure to the sensor on the skin, improving measurement consistency. This combined sensing mechanism of a breathable chamber and actuator allows

**Fig. 5 | Environmental health impact assessment using BSA. a** Schematic illustration of the clinical test assessing particulate matter impact on the skin barrier in an AD patient. **b** Change in MESOR of SH and TEWL over 14 days in an AD patient, according to PM exposure levels. **c** Pearson correlation heatmap showing the relationships and statistical significance between variables such as MESOR of hydration (MH), amplitude of hydration (AH), MESOR of TEWL (MT), amplitude of TEWL (AT), daily mean temperature (DMT), daily mean humidity (DMH), particulate matter exposure (PME), and SCORAD index (SI). **d–f** Scatter plot showing the correlation among MESOR values of SH and TEWL, and PM exposure. **g** Bar graphs

of correlation analysis between PM exposure and SH and TEWL, when before intervention PM exposure ranges from 0–40 and 40–80 mg/m³day. **h** Schematic illustration of interventions to reduce particulate matter exposure in daily life, along with a graph comparing exposure levels during a clinical period. **i–k** Scatter plot showing the correlation among MESOR values of SH and TEWL, PM exposure, and SCORAD Index after intervention. **l** Bar graphs of correlation analysis between PM exposure and SH and TEWL, when after intervention, PM exposure ranges from 0–40 and 40–80 mg/m³day. Statistical significance was assessed using two-sided Pearson's correlation analysis. (**c, d–f, i–k**).

accurate, long-term monitoring of TEWL and SH even during daily activities. Unlike traditional short-term measurements (1–4 times per day), our device provides a more precise assessment of skin condition with long-term measurement data (~ 200 times per day) by considering circadian rhythm over extended periods. The compact watch-sized design and robustness of the device allow for comfortable wear for weeks in everyday life.

We have also demonstrated the capability to quantitatively analyze skin barrier health diagnostics and the environmental health impacts on atopic dermatitis. The actuator and sensor operate according to scenarios that allow sufficient time for TEWL and hydration measurement and ventilation. This autonomous long-term monitoring system is convenient, requiring no complex protocols or training, and can be easily turned on/off and monitored via a smartphone through Bluetooth communication.

This study has limitations in accurately assessing the correlation between environmental factors and AD due to insufficient consideration of individual genetic factors. Genetic predispositions play an important role in the onset and progression of AD, and ignoring these in favor of solely analyzing environmental influences may distort the interpretation of results. Future research should adopt an integrated approach that considers a range of factors, including individual genetic backgrounds.

There are still opportunities for the following improvements to our long-term monitoring device: (i) Miniaturization of the device targeted for infancy and children, (ii) Integration of additional sensors to measure external environmental factors that can affect the skin (e.g., particulate matter, nitrogen dioxide, formaldehyde), (iii) Further clinical studies to quantitatively analyze factors affecting the skin over the long term, (iv) Integrating wearable devices with non-invasive measurement methods (e.g., diffuse reflectance spectroscopy) to minimize damage to patient skin, (v) Development of a device for measuring additional skin health biomarkers (e.g., epidermal serine concentration[57], sweat pH[58]). We envision that this long-term skin health monitoring system will enable individuals with sensitive skin to provide moisture or block external environmental factors based on quantitative indicators, thereby improving skin health care. Moreover, it could be used to identify factors that have long-term adverse effects on the skin barrier, which current technology has yet to elucidate.

## Methods
### FEA simulation
FEA analysis using ABAQUS was conducted to simulate the mechanical deformation and stress distribution in the breathable chamber with PDMS, along with skin composed of the stratum corneum, epidermis, dermis, fat, and muscle[59]. The analysis addressed the mechanical response during the pressing of the top of the chamber. C3D8R (continuum, 3-D, 8-node, reduced integration) elements were used for modeling the chamber, and C3D10 (continuum, 3-D, 10-node) for the skin. The FEA was processed in two steps: (1) mechanical behavior of breathable chamber with a fixed bottom plane under pressing, and (2) analyzing the coupled deformation of the chamber and skin. Both the chamber and skin were modeled as elastic materials characterized by Young's modulus ($E$), Poisson's ratio ($v$), and thickness ($t$), where $E_{PDMS} = 1.4\,MPa$, $v_{PDMS} = 0.49$, $E_{stratum} = 1998$ MPa, $v_{stratum} = 0.48$,

$E_{epidermis} = 102\,MPa$, $v_{epidermis} = 0.48$, $E_{dermis} = 10.2\,MPa$, $v_{dermis} = 0.48$, $E_{fat} = 0.102\,MPa$, $v_{fat} = 0.48$, $E_{muscle} = 888\,MPa$, $v_{muscle} = 0.3$.

The simulation was also conducted to analyze the strain distribution in the sensor components under a maximum load (Supplementary Fig. 12). To set an upper limit on the load the sensor can experience, we consider the critical force of the bistable structure that is connected to the sensor. The sensor is only allowed to experience more than the beam's critical force, which is measured 1.2 N in the experiment. The sensor components and skin were modeled using CPS4R elements (A 4-node bilinear plane stress quadrilateral, reduced integration, hourglass control). Both the elastomer covering the sensor module and the skin were assumed to be linear elastic materials. There mechanical properties, Young's modulus ($E$), and Poisson's ratio ($v$), are summarized below: $E_{elastomer} = 1.4\,MPa$, $v_{elastomer} = 0.49$, $E_{NTC} = 150\,GPa$, $v_{NTC} = 0.3$, $E_{copper} = 130\,GPa$, $v_{copper} = 0.33$, $E_{PI} = 2.5\,GPa$, $v_{PI} = 0.33$, $E_{heater} = 150\,GPa$, $v_{heater} = 0.3$, $E_{epidermal} = 102\,MPa$, $v_{epidermal} = 0.48$, $E_{dermis} = 10.2\,MPa$, $v_{dermis} = 0.48$.

### Natural evaporation
The natural evaporation simulation was conducted using STAR-CCM+ with 3D multiphase interaction and volume of fluid (VOF) analysis. The primary objective was to compare evaporation dynamics between a closed chamber and an open chamber. Both chambers were modeled with water at the bottom and air at the top, with variations in the chamber hole size used as a parameter to study changes in relative humidity during evaporation. A gap of 0.05 mm was maintained between the target areas of both chambers to facilitate the analysis.

The simulation assumed that the air consisted solely of "dry air" and "water vapor," following the ideal gas law. Initial conditions were set at a temperature of 25 °C and a relative humidity of 20%. The gas species included $H_2O$ (gas) and air, with molecular weights of 18.015 g/mol and 28.96 g/mol, respectively. The initial mass fraction of the gas species was defined as $H_2O$ (gas): Air = 0.0043: 0.9957, corresponding to 20% relative humidity $\left(100 \times \frac{P_P}{P_S}\right)$, where $P_P$ and $Ps$ are the partial pressure of $H_2O$ (gas) and saturation vapor pressure. The liquid species was $H_2O$ (liquid). The field functions in the simulation were defined as follows:

$$P_P = H \times \frac{P_U}{\left(\frac{18}{28.96}\right) + H}$$

where $P_U$ is the absolute pressure, and $H$ is the mass fraction ratio of $H_2O$ (gas) to air.

$$P_S = 611.21 \times \exp\left(\frac{18.678 - T/234.5 \times T}{257.14 + T}\right)$$

where $T$ is the temperature of air.

### Actuator fabrication
We first insert a 0.1 mm diameter SMA wire through the holes of two interlocking joints and press two crimps at both ends of the wire (Supplementary Fig. 9a). These crimps serve as electric connections and as mechanical anchors that fix the wire length. The length of the wire is determined based on the length of the beam. There is a slit in the interlocking joint that allows the bistable CFRP beam (0.25 mm

thickness) to fit in (Supplementary Fig. 9b). One set of SMA wire is mounted on the upper side of the beam and the other on the lower side, enabling bidirectional (upward and downward) actuation (Supplementary Fig. 9c). The beam is fabricated slightly longer than the spacing of the aluminum frame into which it is inserted (Supplementary Fig. 9d). This length difference induces the bending of the beam. We bend the beam to fit it into the frame (Supplementary Fig. 9e). Upon activation of either the upper or lower SMA wire, the beam switches its bistable state, bending either upward or downward (Supplementary Fig. 9f).

### Sensor fabrication
The sensor is made in nine steps, as shown in Supplemental Fig. 3. The FPCB containing the SH and TEWL sensors is attached to the 3D printed cap (a). The FPCB includes a pair of resistance heating elements (ERJ-3EKF1000V), two pairs of NTC thermistors (NCP03XH103J05RL), and a commercial humidity sensor (SHT40-AD1B-R2). The polymer layer that contacts the skin is made in two steps. The first layer is a polymer mix of Eco-flex 0030 and Eco-flex gel in a 1:1 ratio, which is poured onto the acryl and spin-coated at 800 rpm for 30 seconds (b). The evenly coated polymer layer is semi-cured for 4 minutes at 70 °C (c). The second layer is for attaching the polymer to FPCB. Silicon adhesive is poured over the first layer and spin-coated at 500 rpm for 30 seconds (d). The evenly coated silicon adhesive is semi-cured for 4 minutes at 120°C (e). Silicon adhesive is also coated in the same way and semi-cured. Adjust the height of the acrylic structure so that only the FPCB with the sensor fabricated in (a) is attached to the second layer, and allow it to cure completely (f). A mold is made on a 3D printer to fit the shape of the chamber, which consists of an upper and lower part. Dragon skin 0030 is poured into the mold and fully cured to produce the chamber (g). The sensor part made in step (f) is attached to the chamber (h). Finally, attach the two sensor parts, one with the N pole upward and the other with the S pole upward (i).

### Information for clinical test
All participants in the study were volunteers and submitted informed consent before the clinical test. A summary of baseline characteristics of the participants is presented in Supplementary Table 9, including information on age, sex, ethnicity, and pathology status. Clinical trial of BSA was conducted under the Institutional Review Board (IRB) protocol (SKUIRB-2023-01-054) approved by Seokyeong University. To monitor skin condition, three healthy individuals and three patients with atopic dermatitis, all from a young age group (23 to 31 years), participated in the test (Fig. 4j–l). The pathological symptoms included xerosis across the entire skin and the presence of erythemaous lesions, which allowed for the clinical diagnosis of atopic symptoms. All participants maintained their usual daily activities and conditions during the test. The selected measurement sites, including healthy areas and regions adjacent to lesions, were gently cleansed with an alcohol swab. The BSA was then positioned on the forearm at these sites, and the device was secured in place using a smartwatch strap. SH and TEWL measurements were monitored via Bluetooth connection between the device and a smartphone. To ensure stable operation of the device, the scenario was designed to initiate the actuator and begin measurements 5 s after establishing the Bluetooth connection. Each participant continued measurements at 5 min intervals during daily activities, pausing only in situations where measurement was impractical, such as during showering or exercise. The continuously collected data over extended periods was stored on a computer web server and used for further data analysis (Supplementary Fig. 2).

### Ethics
Each participant gave informed written consent.

### Reporting summary
Further information on research design is available in the Nature Portfolio Reporting Summary linked to this article.

## Data availability
All data supporting the findings of this study are available within the article and its supplementary files. Any additional requests for information can be directed to, and will be fulfilled by the corresponding authors. Source data are provided with this paper.

## Code availability
Codes for outlier classification used in this study are available on GitHub (https://github.com/Myungrae/BSA.git, https://doi.org/10.5281/zenodo.16608927).

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

## Acknowledgements

This work was supported by the Korea Environment Industry & Technology Institute (KEITI) through the Digital Infrastructure Building Project

for Monitoring, Surveying, and Evaluating the Environmental Health Program, funded by the Korea Ministry of Environment (MOE) (2021003330009). This work was also supported by the Korea Environment Industry & Technology Institute (KEITI) through the Environmental Health Digital Program funded by the Korea Ministry of Environment (MOE) (2021003330010). This work was also supported by the Environmental Health Action Program of Korea Environmental Industry and Technology Institute (KEITI) (Grant no. 2021003380004).

## Author contributions

I.Hong, D.Lim, D.Kim, M.R.Hong, S.Seo, J.-S. Koh, S.Han, and D.Kang conceived the idea and designed the research. I. Hong, D.Lim, C.Im, and J.Lee designed the hardware for the wireless electronic system. D.Kim and T.Oh designed and fabricated the actuator. M.R.Hong and K.Ji analyzed the skin condition data. S.Kang and J.Lee performed software design and validation. S.Hwang and Y.Roh fabricated the sensor. D.Gong and G.Kwon conducted mechanical modeling. I. Hong and D.Lim performed research and led the experimental works with support from T.Kim, E.Kim, S.Kim, J.Kim, Se. Kim and K.Shim. I.Hong, D.Lim, D.Kim, and M.R.Hong performed figure design and writing of the manuscript. S.Seo, J.-S. Koh, S.Han, and D.Kang supervised the research.

## Competing interests

The authors declare no competing interests.
