## [Transparent Peer Review file · Nature Communications]

Breathable, wearable skin analyzer for reliable long-term monitoring of skin barrier function and individual environmental health impacts

Corresponding Author: Professor Daeshik Kang

Version 0:

Reviewer comments:

Reviewer #1

(Remarks to the Author)

Thanks for inviting me to evaluate the manuscript entitled “Breathable skin health analyzer for reliable long-term monitoring of individual environmental health” for Nature Communications. Hong et al. presented a wearable system for TEWL and SH monitoring through a reasonable and impressive device structural design and optimization. Sufficient experiments were carried out to prove that the system has detection accuracy and good anti-interference ability. This manuscript solves the challenge that existing devices in this field cannot monitor skin health indicators for a long time, and demonstrates a suitable and interesting application scenario in personalized environmental health management. After careful review, I think this work can be accepted with the following revisions:

1. According to the contents in this manuscript, it seems that the title is not very suitable. I think one detail missing from the current title is that this system was used for skin barrier health monitoring. The original intention of the measurement of TEWL and SH levels is to evaluate the individual barrier health.
2. The introduction does not fully summarize the advances in the field of existing wearable devices for TEWL and SH measurement. Therefore, the literature review is insufficient and unsolid, and this manuscript is not well presented by highlighting the advantages of this innovation over other works. The authors can provide effective support for the novelty of their work by reading and summarizing the following key literatures in the field:
 - a. Ventilated chamber method: Sivakumar A. D., Sharma R., Thota C., Ding D. & Fan X. WASP: wearable analytical skin probe for dynamic monitoring of transepidermal water loss. *ACS Sens.* 8, 4407–4416 (2023).
 - b. Open chamber method: Salvo P., Pingitore A., Barbini A. & Di Francesco F. A wearable sweat rate sensor to monitor the athletes' performance during training. *Sci. Sports* 33, e51–e58 (2018).
 - c. Open chamber method: Liu Y., et al. Skin-interfaced superhydrophobic insensible sweat sensors for evaluating body thermoregulation and skin barrier Functions. *ACS Nano* 17, 5588–5599 (2023).
 - d. Review: Zhong B., Jiang K., Wang L. Wearable sweat loss measuring devices: from the role of sweat loss to advanced mechanisms and designs. *Adv. Sci.* 9, 2103257 (2022).
3. The authors compare the performance of the system in this work with other commercial and existing wearable electronics in Supplementary Tables 1 and 2, which include a lot of reference numbers (Ref) that cannot be found in the supplementary materials or anywhere else.
4. To validate the measurement accuracy and repeatability of the SH sensor based on thermal sensing, practical experimental tests and comparisons based on standard materials with known thermal characteristics are necessary. The authors can refer to the experimental scheme proposed by Madhvapathy, et al. (Madhvapathy S. R., et al. Reliable, low-cost, fully integrated hydration sensors for monitoring and diagnosis of inflammatory skin diseases in any environment. *Sci. Adv.* 6, eabd7146 (2020).)
5. For the TEWL sensor, the calibration curve of TEWL levels versus humidity rising rates is required to quantitatively evaluate the sensor performance, including linearity, sensitivity, detection range, and limit of detection (LOD).
6. Will changes in skin temperature affect the measurements of the two sensors?
7. The baseline characteristics of participants should be provided in this work.
8. Are there statistically significant differences in the two skin indicators between non-lesional sites near the lesional skin of AD patients and normal sites of healthy participants?

9. Long-term monitoring is indeed not friendly to the diseased skin of AD patients. So, I suggest improving the test method to prove its effectiveness for skin disease detection, such as using the analyzer to perform a single, short-term measurement on the lesional skin.

Reviewer #2

(Remarks to the Author)

this work propose a wearable device with size comparable to that of a smartwatch.

The device aims to measure the skin humidity (SH) and transdermal water loss (TEWL). this is achieved by a micromechanical system (MEMS) that includes thermistors and heater for SH, a breathable chamber and a humidity sensor for TEWL. the work includes a 28-day validation on three patients. The complexity and the originality of the device's design and the clinical validation may be worth of publication. The reviewer has, however, some comments which may clarify some points and improve the paper. Briefly:

1. The manuscript is well written however additional details on the working principle of the sensors should be included in the main text (part of note 1 and 2 for example);

2. the MEMs operating principle is not totally clear to the reviewer. the role of SMA should be better explain. why SMA wires are needed? what is the physical mechanism behind the operation?

3. Fig1.b shows a magnet and a cap. the reviewer misses the explanation of their functions.

4. the complexity of hte micromechanical system highlight the ability of the involved research groups to engineer and craft wearable devices. however, complexity also raises concerns about reliability especially for wearable-on skin MEMS devices where sweat, movements (friction etc..) may bring to failure. can the authors comment on this? did they run any simulation to evaluate the stress arising during operation (snapping and release)?

5. the authors do not comment on the power consumption of the device. this would be something interesteing for the readers to know.

7. methods section should include the exact models of the chip used (NTC, heaters, etc..). is the humidity sensors a commercial one?

Version 1:

Reviewer comments:

Reviewer #1

(Remarks to the Author)

The authors have satisfactorily addressed all my concerns. However, I think a recent work on skin health monitoring published in Nature Communications (An Epidermal Serine Sensing System for Skin Healthcare, Nature Communications, 2025, 16, 2681.) should be cited and If the author wants to claim the system's key advantage, they should analyzed for its differences from this work.

Reviewer #2

(Remarks to the Author)

The authors have addressed all comments and responded to all questions from the reviewers. It can be accepted as it is.

Reviewer: 1

Thanks for inviting me to evaluate the manuscript entitled “Breathable skin health analyzer for reliable long-term monitoring of individual environmental health” for Nature Communications. Hong et al. presented a wearable system for TEWL and SH monitoring through a reasonable and impressive device structural design and optimization. Sufficient experiments were carried out to prove that the system has detection accuracy and good anti-interference ability. This manuscript solves the challenge that existing devices in this field cannot monitor skin health indicators for a long time, and demonstrates a suitable and interesting application scenario in personalized environmental health management. After careful review, I think this work can be accepted with the following revisions:

Our response:

We sincerely appreciate the reviewer’s constructive inputs and insightful comments on our manuscript. The suggestions have helped us identify areas to improve both the clarity and completeness of our work. Specifically, we refined the title to emphasize skin-barrier function and enriched the introduction with a more comprehensive review of recent advances in TEWL and SH sensing technologies. We also added new calibration and validation experiments that substantiate the accuracy and repeatability of our sensors. We have carefully addressed each comment and revised the manuscript accordingly.

Comment (1) According to the contents in this manuscript, it seems that the title is not very suitable. I think one detail missing from the current title is that this system was used for skin barrier health monitoring. The original intention of the measurement of TEWL and SH levels is to evaluate the individual barrier health.

Our response:

We are grateful to the reviewer for pointing out that the original title did not explicitly refer to skin barrier function. We agree that the term “skin health” is too broad and does not capture our primary focus on barrier function via TEWL and SH measurements. Accordingly, we have revised the title as “Breathable, wearable skin analyzer for reliable long-term monitoring of skin barrier function and individual environmental health impacts” throughout the manuscript. We believe that this new title more accurately reflects our work’s intended objective and application in skin barrier health monitoring.

Original title: Breathable skin health analyzer for reliable long-term monitoring of individual environmental health

Revised title: Breathable, wearable skin health analyzer for reliable long-term monitoring of skin barrier function and individual environmental health impacts

Our modification to the manuscript:

(Page 1 line 1-2)

Breathable, **wearable** skin **health** analyzer for reliable long-term monitoring of **skin barrier function and** individual environmental health **impacts**

**Breathable, wearable skin analyzer for reliable long-term
monitoring of skin barrier function and individual
environmental health impacts**

Comment (2) The introduction does not fully summarize the advances in the field of existing wearable devices for TEWL and SH measurement. Therefore, the literature review is insufficient and unsolid, and this manuscript is not well presented by highlighting the advantages of this innovation over other works. The authors can provide effective support for the novelty of their work by reading and summarizing the following key literatures in the field:

- a. Ventilated chamber method: Sivakumar A. D., Sharma R., Thota C., Ding D. & Fan X. WASP: wearable analytical skin probe for dynamic monitoring of transepidermal water loss. *ACS Sens.* 8, 4407–4416 (2023).
- b. Open chamber method: Salvo P., Pingitore A., Barbini A. & Di Francesco F. A wearable sweat rate sensor to monitor the athletes' performance during training. *Sci. Sports* 33, e51–e58 (2018). - 9
- c. Open chamber method: Liu Y., et al. Skin-interfaced superhydrophobic insensible sweat sensors for evaluating body thermoregulation and skin barrier Functions. *ACS Nano* 17, 5588–5599 (2023).
- d. Review: Zhong B., Jiang K., Wang L. Wearable sweat loss measuring devices: from the role of sweat loss to advanced mechanisms and designs. *Adv. Sci.* 9, 2103257 (2022).

Our response:

We would like to express our sincere appreciation to the reviewer for providing valuable feedback. In response, we have thoroughly reviewed the suggested paper along with other relevant literature and incorporated the recent advancements in TEWL and SH measurement technologies into the introduction section of the manuscript. According to the references, the open-chamber method for measuring TEWL allows for long-term measurements by not covering the skin, but it is vulnerable to environmental disturbances such as ambient airflow. In contrast, the closed-chamber method is less affected by external factors and provides higher measurement accuracy but requires regular removal of accumulated water vapor.

Our device is designed to combine the high measurement accuracy of a closed chamber with the ventilation capability of an open chamber. During measurement, the chamber is brought into contact with the skin, and during ventilation, it is lifted off the skin. To achieve

this, we employed an actuator composed of SMA (Shape Memory Alloy) and a bistable beam. Compared to previously developed devices, our system is more compact and energy-efficient, enabling stable, long-term measurement of TEWL and SH. By integrating a robust measurement mechanism against external noise, a ventilation system that enables long-term measurement, and a wireless communication-based monitoring system into a compact form factor, we have developed a wearable device capable of long-term monitoring of physiological data. We have revised the manuscript to better highlight the innovative advantages and contributions of our work.

Our modification to the manuscript:

(Page 4 line 66-84)

~~To address these issues, breathable sensors²²⁻²⁴ with ultrathin substrates with nano-mesh and micro-hole patterns have been developed. While these sensors improve sweat evaporation, they remain vulnerable to mechanical friction and environmental noise, limiting their suitability for everyday use by the general public^{23,24} (Supplementary Table 2). Another approach to avoid sweat accumulation includes integrating actuators to enable the sensor detachment for ventilation. However, the bulky size or tethered system limit their use for wireless, comfortable, and suitable everyday applications^{25,26}. To address these issues, breathable sensors²²⁻²⁴ with ultrathin substrates with nano-mesh^{25,26} and micro-hole patterns have been developed. While these sensors improve sweat evaporation, they remain vulnerable to mechanical friction and environmental noise, limiting their suitability for everyday use by the general public^{23,24,27}. Another approach to prevent sweat accumulation is to design a chamber that allows ventilation at the skin surface²⁸. Open-chamber systems allow natural ventilation by exposing the skin to ambient air; however, they are highly susceptible to environmental disturbances such as airflow, which compromises measurement accuracy under daily conditions^{29,30}. In contrast, closed-chamber systems offer improved measurement~~

accuracy by isolating the skin from ambient airflows, but they require a mechanism to regularly remove accumulated moisture, leading to increased device bulk and limited applicability for wearable devices³¹⁻³³ (Supplementary Table 2). Alternatively, conventional epidemiologic studies have relied on survey-based skin health scoring methods, such as the SCORAD index for atopic dermatitis^{27,28}. These methods are limited by subjective factors including itching and sleep disturbances, which can reduce the consistency and accuracy of the data as they depend on individual emotional responses or experiences^{29,30}. To develop a wearable device capable of long-term monitoring, it is essential to miniaturize the chamber and actuator to prevent sweat accumulation, enable monitoring through wireless communication, ensure low power consumption for extended use, and incorporate the ability to measure objective indicators of the skin.

(Reference)

- 26 Zhong, B., Jiang, K., Wang, L. & Shen, G. Wearable sweat loss measuring devices: from the role of sweat loss to advanced mechanisms and designs. *Advanced Science* **9**, 2103257 (2022).
- 27 Liu, Y. *et al.* Skin-interfaced superhydrophobic insensible sweat sensors for evaluating body thermoregulation and skin barrier functions. *ACS nano* **17**, 5588-5599 (2023).
- 28 Salvo, P., Pingitore, A., Barbini, A. & Di Francesco, F. A wearable sweat rate sensor to monitor the athletes' performance during training. *Science & Sports* **33**, e51-e58 (2018).
- 29 Matsui, N. *et al.* Wearable System for Continuous Estimation of Transepidermal Water Loss. *Electronics* **13**, 4779 (2024).
- 30 Sim, J. K., Yoon, S. & Cho, Y.-H. Wearable sweat rate sensors for human thermal comfort monitoring. *Scientific reports* **8**, 1181 (2018).
- 31 Sivakumar, A. D., Sharma, R., Thota, C., Ding, D. & Fan, X. WASP: Wearable Analytical Skin Probe for Dynamic Monitoring of Transepidermal Water Loss. *ACS sensors* **8**, 4407-4416 (2023).

Comment (3) The authors compare the performance of the system in this work with other commercial and existing wearable electronics in Supplementary Tables 1 and 2, which include a lot of reference numbers (Ref) that cannot be found in the supplementary materials or anywhere else.

Our response:

We thank the reviewer for the valuable comment. In response to Comment 2, we have revised Supplementary Table 2 to compare the prior studies cited in the introduction with our work. Supplementary Table 1, on the other hand, summarizes the performance of commercial devices; since these are product-based comparisons, there are no direct references available.

Briefly, most commercial devices are designed in a stick-type wearable form, making regular and long-term measurements difficult. Skin hydration is typically measured based on electrical properties such as capacitance (e.g., Corneometer, Scalar, Epsilon), conductance (e.g., DermaLab, Skicon), or impedance (e.g., NOVA, GPskin). These methods are highly susceptible to external factors like cosmetics or lotions, making them unsuitable for everyday, long-term use.

TEWL is commonly measured using either an open-chamber method (e.g., DermaLab, Tewameter) or a closed-chamber method (e.g., GPskin, AquaFlux). The open-chamber method is vulnerable to environmental interference, while the closed-chamber method requires detachment from the skin for ventilation, making it unsuitable for wearable applications.

Our modification to the manuscript:

(Supplementary table 2)

Ref	Research Description	Disadvantages	Device Size	Breathability	Tethered / Untethered
Our Work	Heat transfer-based skin hydration measurement Humidity sensor-based TEWL measurement Chamber ventilation system for repetitive measurements Bluetooth communication	-	~38.5mm x 46mm	98.6%	Untethered
[19]	Heat transfer-based skin hydration measurement	Low breathability Not available for long term measurement	~25mm x 40mm	-	Untethered
[22]	Highly breathable using nano-mesh electrodes Capacitance based skin hydration measurement	Vulnerable to environment noise Low durability	~28mm x 20mm	-	Tethered
[24]	Highly breathable using micro holes Capacitance based skin hydration measurement	Vulnerable to environment noise Low durability	~0.14mm x 0.22mm	94.54%	Tethered
[18]	Heat transfer-based skin hydration measurement Highly resistant to environment noise	Low breathability	~0.9mm x 2.6mm	-	Untethered
[20]	Heat transfer-based skin hydration measurement Highly resistant to environment noise	Low breathability	~25mm x 35mm	-	Untethered
[27]	Skin attachable sensor Capacitance based skin hydration measurement	Vulnerable to environment noise Low durability Low breathability	~1.5mm x 1.5mm	-	Tethered
[26]	Conductivity based TEWL measurement Highly breathable using nano-mesh electrodes	Vulnerable to environment noise Low durability	~14mm x 24mm (electrode) 16mm x 23mm (module)	-	Untethered
[33]	Humidity sensor-based TEWL measurement Chamber ventilation system for repetitive measurements	Low wearability Low breathability	89.9mm x 40.9mm	-	Untethered

[30]	Humidity sensor-based TEWL measurement	Vulnerable to environmental noise	15mm x 15mm	-	Untethered
	Open chamber ventilation system	Low wearability			
[29]	Resistance changes in the composite material based TEWL measurement	Relatively low breathability than open chamber	25mm x 30mm	66.68%	Untethered
	Highly breathable using hole pattern	Low wearability			
[31]	Humidity sensor-based TEWL measurement	Vulnerable to environmental noise	~ 80mm x 80mm	-	Untethered
	Open chamber ventilation system	Low wearability			

Supplementary Table 2 | Performance Comparison with wearable electronics

Comment (4) To validate the measurement accuracy and repeatability of the SH sensor based on thermal sensing, practical experimental tests and comparisons based on standard materials with known thermal characteristics are necessary. The authors can refer to the experimental scheme proposed by Madhvapathy, et al. (Madhvapathy S. R., et al. Reliable, low-cost, fully integrated hydration sensors for monitoring and diagnosis of inflammatory skin diseases in any environment. *Sci. Adv.* 6, eabd7146 (2020).)

Our response:

We thank the reviewer for emphasizing the need to validate accuracy and repeatability of skin hydration sensors. As the reviewer pointed out, for long-term operation and clinical application, not only measurement accuracy and repeatability but also calibration to minimize performance variation between sensors was critically considered in our study. Based on the reference provided by the reviewer [S. Madhvapathy et al., *Science Advances* 6, eabd7146, 2020], a two-point calibration was performed using two elastomeric materials (sylgard 170, sylgard 184). Unlike the calibration method of reference that relies on one point calibration using water, which only allows for offset (intercept) correction, we employed the two-point calibration approach using two types of elastomer materials with known thermal characteristics. This approach enables accurate calibration of both the offset and the slope of sensor response.

To describe this process in detail, as shown in Supplementary Table 6, the thermal properties of Sylgard 170, Sylgard 184, thermal conductivity, thermal diffusivity, density, and specific heat capacity were measured using the LFA 467 HyperFlash instrument. Based on these measured values, simulations were conducted using Sylgard 170 and Sylgard 184 as the substrate materials, as illustrated in Fig. L1.

The principle of the calibration of skin hydration sensor is based on transient heat transfer, and the linear governing relationship between skin hydration (ϕ_s) and the temperature difference (ΔT) between two pairs of NTC thermistors was derived through finite element analysis (FEA) (Supplementary Fig. 7a). Depending on the heating duration—2 seconds and 10 seconds—heat transfer into the skin to depths corresponding to the epidermis and dermis, respectively. The resulting temperature differences enable the estimation of epidermal hydration (ϕ_E) and dermal hydration (ϕ_D) (Supplementary Fig. 7b). As skin hydration increases, the temperature difference decreases. Simulations were conducted at hydration levels ranging from 0% to 100% in 10% increments, and the results confirmed a strong linear correlation between hydration level and temperature difference (Supplementary Fig. 7c,d).

We performed a two-point calibration using S170 and S184 whose thermal conductivities span the expected range of skin hydration. Through sensor calibration, we were able to improve not only the performance of individual sensors but also ensure consistency across different sensors. The results showed high accuracy and repeatability, with the error between the mean and true value of each sensor being less than 2% and the standard deviation of each sensor being less than 5% when the three sensors were tested 10 times. (Supplementary Fig. 7e).

Figure L1 | Measurement Principle and Performance Analysis of the Skin Hydration (SH) Sensor. **a**, Transient heat transfer simulation results at the epidermis (2 s) and dermis (10 s) for calibration materials (sylgard 170, sylgard 184). **b**, Correlation between heating time and the temperature difference measured by the NTC. **c**, Correlation between dermal hydration (Φ_D) and NTC temperature difference with linear fitting results. **d**, Correlation between epidermal hydration (Φ_E) and NTC temperature difference with linear fitting results. **e**, Measurement accuracy and repeatability of skin hydration based on calibration results. Measurements of SH for three BSA sensors are highly accurate and repeatable.

Sensor measurement for three BSA sensors

Measurement	Sensor1	Sensor2	Sensor3
	-15.1356	-12.3817	-15.5993
	-17.0645	-13.2811	-14.221
	-17.1963	-12.7265	-16.8435
Sylgard 184 (SH)	-15.0398	-15.3047	-14.7333
	-17.4479	-12.2917	-15.1358
True value:	-17.1963	-12.6215	-13.3428
-14.9	-13.8896	-15.1548	-15.2822
	-13.65	-16.1142	-15.8067
	-13.9375	-16.1292	-13.0866
	-14.7882	-15.2897	-14.9285
Mean value ± error	-15.5346 ± 1.9	-14.1295 ± 2.0	-14.8980 ± 2.0
	47.05094	45.0149	46.4125
	44.5350	45.1498	43.8022
	45.9847	45.8244	46.1319
Sylgard 170 (SH)	47.5302	43.4710	43.2290
	45.4456	45.6895	42.9484
True value:	45.8888	43.0812	43.0704
44.9	45.8049	43.7558	42.5825
	45.4335	44.5802	45.3147
	46.5957	43.5909	45.5587
	45.4575	47.1735	45.3513
Mean value ± error	45.9727 ± 1.6	44.7331 ± 2.4	44.4402 ± 2.0

Our modification to the manuscript:

(Page 8 line 156-159)

Detailed information of processes to calculate moisture content (ϕ_s) is provided in Supplementary Note 1. Figure 2c shows a finite element analysis (FEA) demonstrating how the temperature distribution changes with SH level upon heating (Supplementary Fig. 6 and Supplementary Table 6).

(Supplementary Figure 7.)

Supplementary Figure 7 | Measurement Principle and Performance Analysis of the Skin Hydration (SH) Sensor. **a**, Transient heat transfer simulation results at the epidermis (2 s) and dermis (10 s) for calibration materials (sylgard 170, sylgard 184). **b**, Correlation between heating time and the temperature difference measured by the NTC. **c**, Correlation between dermal hydration (Φ_D) and NTC temperature difference with linear fitting results. **d**, Correlation between epidermal hydration (Φ_E) and NTC temperature difference with linear fitting results. **e**, Measurement accuracy and repeatability of skin hydration based on calibration results. Measurements of SH for three BSA sensors are high accuracy and repeatability.

(Supplementary Note 1.)

The principle of the calibration of skin hydration sensor is based on transient heat transfer, and the linear governing relationship between skin hydration (ϕ_s) and the temperature difference (ΔT) between two pairs of NTC thermistors was derived through finite element analysis (FEA) (Supplementary Fig. 7). Depending on the heating duration—2 seconds and 10 seconds—heat transfer into the skin to depths corresponding to the epidermis and dermis, respectively. The resulting temperature differences enable the estimation of epidermal hydration (ϕ_E) and dermal hydration (ϕ_D). Two materials (Sylgard 170 and Sylgard 184) were used as calibration standards. Their thermal properties correspond to 34.9% and -7.9% hydration levels in the epidermis, and 44.9% and -14.9% in the dermis, respectively. ~~The skin hydration sensor is calibrated using a material with similar thermal properties to skin (Sylgard 170, Sylgard 184). These materials have thermal properties of 34.9% and -7.9% of epidermal skin hydration when heated for 2 seconds, respectively. In addition, we can measure the skin hydration of the dermis when heated for 10 seconds, and the above materials have thermal properties of 44.9% and -14.9% of dermal skin hydration, respectively.~~

(Supplementary Note 1.)

This equation can be applied to both epidermal and dermal hydration levels to obtain four calibration factors, two each. The calibration factors are obtained by measuring the temperature difference between the skin and a material with similar thermal properties so that the device can reliably provide a wide range of skin hydration values caused by skin diseases. ~~Through sensor calibration, we were able to improve not only the performance of individual sensors but also ensure consistency across different sensors. The results showed high accuracy and repeatability. When the three sensors were tested 10 times, the error between the mean value and true value of each sensor was less than 1%, and the error within each sensor was less than 2%.~~

(Supplementary Table 6.)

Material	Thermal conductivity, k (W/mK)	Thermal diffusivity, α (m²/s)	Density, ρ (kg/m³)	Specific heat capacity, C_p (J/kgK)
Moist skin	0.6	0.14	-	-
Dry skin	0.2	0.15	-	-
Polyimide	0.55	-	1340	3731
Copper	377	-	8940	385
Skin adhesive	0.164	0.12	969	1405
Silver	429	-	10500	234
Alumina	20	-	3900	900
Sylgard 184	0.165	0.11	987	1509
Sylgard 170	0.324	0.208	1327	1177

Supplementary Table 6 | Thermal properties of each material used for transient heat transfer simulation analysis.

Comment (5) For the TEWL sensor, the calibration curve of TEWL levels versus humidity rising rates is required to quantitatively evaluate the sensor performance, including linearity, sensitivity, detection range, and limit of detection (LOD).

Our response:

We thank the reviewer for the valuable suggestion regarding the quantitative evaluation of the TEWL sensor's performance. In response, we conducted experiments to analyze the correlation between the relative humidity rising measured by our TEWL sensor.

TEWL values were obtained using the wet-cup method by systematically varying the water temperature (Figure L2a and b). The humidity rising rate within the closed chamber was measured over a programmed time using our device, as shown in Figure L2c. The resulting calibration curve exhibited excellent linearity, as shown in Figure L2d ($R^2 = 0.9995$). From the analysis, key performance metrics were extracted: the sensor exhibited a sensitivity of 0.1173 (%RH/s) per ($\text{g}\cdot\text{m}^{-2}\cdot\text{h}^{-1}$), a detection range from 5.02 to 37.6 $\text{g}\cdot\text{m}^{-2}\cdot\text{h}^{-1}$, and a limit of detection (LOD) of 0.0326 $\text{g}\cdot\text{m}^{-2}\cdot\text{h}^{-1}$. The LOD was calculated using the standard method, $\text{LOD} = \frac{3.3 \sigma}{S}$, where σ is the standard error of the slope and S is the slope of the calibration curve.

The sensor exhibited excellent linearity, confirming its ability to provide reliable and quantitative measurements within a physiologically relevant range [Montero-Vilchez, T. et al. Skin barrier function in psoriasis and atopic dermatitis: transepidermal water loss and temperature as useful tools to assess disease severity. *J. Clin. Med.* 10, 359 (2021)].

Figure L2. Performance evaluation curves of the TEWL sensor. **a**, Image of the wet-cup method used a hot plate. **b**, Image of the BSA placed on the semi-permeable membranes during measurement. **c**, Relative humidity increase measured in the closed chamber at various TEWL levels induced by different water temperatures using the wet-cup method. **d**, Linear correlation between the humidity rising values measured by the TEWL sensor. All measurements were repeated five times ($n = 5$), and error bars represent standard deviations.

Comment (6) Will changes in skin temperature affect the measurements of the two sensors?

Our response:

We appreciate the reviewer's insightful comment regarding the potential influence of skin temperature on the measurements of both the skin hydration and TEWL sensors.

To address skin temperature variations on the skin hydration sensor, we conducted additional experiments as presented in Supplementary Fig. 5. In these experiments, we systematically varied the ambient temperature (T_a) using an oven and a refrigerator, and the substrate temperature (T_s) using a hot plate and an air blower, while simultaneously measuring the sensor outputs (T_1 , T_2 , and ΔT_{12}). ΔT_{12} represents the differential temperature between two points measured using two pairs of NTC thermistors (NTC1 for T_1 and NTC2 for T_2), and is utilized to minimize the influence of environmental fluctuations during hydration sensing. While the absolute temperature values (T_1 , T_2) can fluctuate due to external factors such as ambient temperature or airflow, the differential temperature (ΔT_{12}) remains relatively insensitive to such variations and predominantly reflects changes associated with local evaporation rates resulting from skin hydration changes. As shown in the results, although T_1 and T_2 varied with changes in T_a and T_s , respectively, ΔT_{12} remained relatively stable throughout the experiments, even under various external conditions (Supplementary Fig. 5a and b) and substrate temperature fluctuations (Supplementary Fig. 5c and d). These findings suggest that our sensor system exhibits low sensitivity to temperature variations and maintains robust performance even against substrate temperature changes, indicating that skin temperature fluctuations are unlikely to significantly affect sensor measurements. This temperature compensation strategy is consistent with approaches reported by the Rogers group [Kwon, K. et al. Wireless, soft electronics for rapid, multisensor measurements of hydration levels in healthy and diseased skin. *Proc. Natl. Acad. Sci. U.S.A.* 118, e2020398118 (2021)], further supporting the reliability of our hydration sensing system under diverse thermal environments.

For the TEWL sensor, it is generally recognized that an increase in skin temperature enhances water diffusion through the stratum corneum, thereby elevating TEWL values [Grice, K. et al. The evaporimetry of human skin. *J. Invest. Dermatol.* 57, 108–110 (1971)]. Accordingly, our sensor is expected to register higher TEWL values under increased skin temperature conditions. To support this, we conducted experiments using the wet-cup method, in which increasing the water temperature led to a higher vapor flux through a semi-permeable

membrane, resulting in increased TEWL sensor outputs as shown Figure L3. Therefore, these findings and previous studies indicate that rises in skin temperature can lead to corresponding increases in TEWL sensor readings.

Additionally, we considered the scenario in which the TEWL value remains constant while only the skin temperature varies. In our device, the humidity sensor used to calculate TEWL is not in direct contact with the skin. It operates within a closed chamber and collects data over approximately 9 seconds following a ventilation period. The measurement strategy is designed to effectively minimize the influence of transient skin temperature changes on the sensor output. Furthermore, according to the manufacturer’s datasheet, the commercial humidity sensor (SHT40-AD1B-R2) used for TEWL measurement maintains a small maximum error of $\pm 3\%$ relative humidity even under high-temperature conditions up to $80\text{ }^{\circ}\text{C}$ [*SHT4x – Humidity and Temperature Sensor Datasheet*. Sensirion AG, 2022. Available at: https://sensirion.com/media/documents/33FD6951/67EB9032/HT_DS_Datasheet_SHT4x_5 (Accessed: 27 May 2025)]. These specifications support the reliability of humidity measurements across a wide range of thermal environments.

In conclusion, the structural design of our system and the characteristics of the TEWL sensor suggest that while skin temperature may physiologically influence actual TEWL, its effect on the measured sensor output is expected to be minimal.

Figure L3. TEWL values measured at different water temperatures.

Our modification to the manuscript:

(Page 7 line 140-164)

The SH sensor includes two pairs of NTC temperature sensors and a pair of heaters. One pair of NTCs (NTC 2) is placed above the heaters, while the other (NTC 1) is located 1.5 mm away to form a temperature gradient reference (Supplementary Fig. 4). When the SH sensor contacts the skin, the SH sensor applies heat to the resistance heating element (heater) and measures the temperature (T_1 :NTC 1, T_2 : NTC 2) at NTCs. ~~The difference between the two temperatures ($\Delta T = T_2 - T_1$) helps eliminate ambient temperature fluctuations, ensuring reliable moisture assessments (Supplementary Fig. 5).~~ ΔT_{12} , defined as the differential temperature between T_1 and T_2 , is relatively insensitive to external factors such as ambient or substrate temperature, and predominantly reflects local changes associated with evaporation rates due to skin hydration levels. The temperature-compensation strategy enhances the robustness of the sensing system against environmental disturbances and skin temperature, as validated through additional experiments presented in Supplementary Fig. 5. Using transient heat transfer, we derive the thermal characteristics of the skin, such as thermal conductivity (k) and thermal diffusivity (α), and from these, the skin's moisture content (ϕ_s). Detailed information of processes to calculate moisture content (ϕ_s) is provided in Supplementary Note 1. Figure 2c shows a finite element analysis (FEA) demonstrating how the temperature distribution changes with SH level upon heating (Supplementary Fig. 6). By adjusting heating time—2 seconds for epidermal hydration (ϕ_E) at ~100 μm depth and 10 seconds for dermal hydration (ϕ_D) at ~1400 μm depth¹⁹—we can analyze both the epidermal layer and the dermal layer to provide a more accurate assessment of skin hydration. Figure 2d presents that the epidermis and dermis allow different ΔT depending on the SH level ranging from 0 to 100 %. The results can derive ϕ_E and ϕ_D by applying the micromechanical modeling that relates the temperature sensor to SH.

Supplementary Fig. 5 | Elimination of temperature fluctuations using NTCs under various conditions. **a**, Measurements of T_1 (yellow), T_2 (red) and ΔT (blue) in various ambient temperature (T_a , black) using an oven and a refrigerator (red and blue background, respectively), **b**, Low sensitivity of ΔT_{12} to change in T_a through temperature compensation ($\Delta T_1 - \Delta T_2$). **c**, Measurements of T_1 (yellow), T_2 (red) and ΔT (blue) in various substrate temperature (T_s , black) using a hot plate and an air pump (red and purple background, respectively), **d**, Low sensitivity of ΔT_{12} to change in T_s through temperature compensation.

Comment (7) The baseline characteristics of participants should be provided in this work.

Our response:

We appreciate the reviewer’s valuable comment regarding the importance of presenting the baseline characteristics of the study participants. In response, we have included the Supplementary Table 9 summarizing the number of subjects, age, sex, ethnicity, and pathology status. We believe this addition enhances the clarity and completeness of the description of the study cohort.

Our modification to the manuscript:

(Supplementary Table 9)

Subject /Number	Age	Sex	Ethnicity	Pathology
Normal 1	33	M	Asian	Healthy
Normal 2	29	F	Asian	Healthy
Normal 3	27	M	Asian	Healthy
AD 1	26	M	Asian	Atopic dermatitis
AD 2	29	F	Asian	Atopic dermatitis
AD 3	24	M	Asian	Atopic dermatitis

Supplementary Table 9 | Baseline characteristics of the participants enrolled in this study.

(Page 24 line 574-593)

Information for clinical test.

All participants for the study were voluntary and submitted the informed consent before the clinical test. A summary of baseline characteristics of the participants is presented in Supplementary Table 9, including information on age, sex, ethnicity, and pathology status. The clinical trial of BSA was conducted under the Institutional Review Board (IRB) protocol (SKUIRB-2023-01-054) approved by Seokyeong University. To monitor skin condition, three healthy individuals and three patients with atopic dermatitis, all from a young age group, participated in the test (Fig. 4j-l). The pathological symptoms included xerosis across the entire skin and the presence of erythematous lesions, which allowed for the clinical diagnosis of atopic symptoms. All participants maintained their usual daily activities and conditions during the test. The selected measurement sites, including healthy areas and regions adjacent to lesions, were gently cleansed with an alcohol swab. The BSA was then positioned on the forearm at these sites, and the device was secured in place using a smart watch strap. SH and TEWL measurements were monitored via Bluetooth connection between the device and a smartphone. To ensure stable operation of the device, the scenario was designed to initiate the actuator and begin measurements 5 s after establishing the Bluetooth connection. Each participant continued measurements at 5 min intervals during daily activities, pausing only in situations where measurement was impractical, such as during showering or exercise. The continuously collected data over extended periods were stored on a computer web server and used for further data analysis (Supplementary Fig. 2).

Comment (8) Are there statistically significant differences in the two skin indicators between non-lesional sites near the lesional skin of AD patients and normal sites of healthy participants?

Our response:

We thank the reviewer for the valuable comment regarding the statistical comparison between non-lesional sites in AD patients and normal sites in healthy participants. To address the question, we first provide a brief summary of our clinical study, followed by additional analyses of the statistical differences between the two groups.

In our study, BSA was demonstrated on three AD patients and three healthy individuals over a three-day period. To avoid worsening symptoms, the device was worn on non-lesional sites of the forearm near lesions, following prior studies showing consistent SH and TEWL trends in such areas [S. Seidenari et al., *Acta dermato-venereologica*, 75. 429, 1995], [B. Eberlein-König et al., *Acta dermato-venereologica*, 80, 188, 2000], [F. Addor et al., *International journal of dermatology*, 51, 672, 2012]. Hourly-averaged data were analyzed to assess statistical differences between groups. Rather than relying on single SH or TEWL values, we focused on their variation patterns and the correlation between the two parameters, which enabled clear differentiation between AD and healthy subjects.

As suggested by the reviewer, we performed statistical comparisons between groups using a t-test and an F-test to assess differences in the means and variances of SH and TEWL, respectively (Figure L4a). For skin hydration, the mean values between the groups were similar, but the AD group showed greater variance, with statistical significance observed only in the variance. Interestingly, the average SH in the AD group appeared slightly higher, which may be attributed to the relatively low SH value of Normal 1 that lowered the overall mean of the control group, as shown in Figure L4b. To investigate this, we reviewed prior literature and found that in Reference Table L1 [S. Seidenari et al., *Acta dermato-venereologica*, 75. 429, 1995], the uninvolved skin of AD patients on the volar forearm showed a higher mean SH compared to healthy controls, but with a larger standard deviation.

In the case of TEWL, both the mean and variance were higher in the AD group compared to the healthy control group, and both differences were statistically significant. This finding is consistent with the results shown in Reference Table L2 [S. Seidenari et al., *Acta dermato-venereologica*, 75. 429, 1995], where the uninvolved skin of AD patients on the volar forearm also exhibited higher mean TEWL and standard deviation than healthy control subjects, with

statistically significant differences between the groups. These results support the reliability of our dual-parameter monitoring approach and demonstrate the importance of long-term skin barrier monitoring for the quantitative diagnosis of AD.

Figure L4. Three-day continuous monitoring of skin barrier function using the BSA in three AD patients and three healthy controls. a, Violin plots of SH and TEWL data from 3 AD patients and 3 healthy controls with statistical comparisons using t-test (mean) and F-test (variance). **b,** Violin plots of SH and TEWL measurements from individual participants (AD1–3, Normal1–3), highlighting inter-individual variability within each group.

[FIGURE REDACTED]

Reference Table L1. Skin hydration (Capacitance values) in children affected by AD and in control subjects (mean \pm SD). The number of assessed areas is in parenthesis.

[FIGURE REDACTED]

Reference Table L2. TEWL values (g/m²h) in children affected by AD and in control subjects (mean \pm SD). The number of assessed areas is in parenthesis.

Our modification to the manuscript:

(Page 16 line 373-375)

Statistical comparisons between non-lesional skin of AD patients and healthy skin, along with the correlation analysis for each individual in the clinical and control groups ~~is~~, are detailed in Supplementary Fig. 19.

(Supporting information)

Supplementary Fig. 19 | Three-day continuous monitoring of skin barrier function using the BSA in three AD patients and three healthy controls. a, Violin plots of SH and TEWL data from 3 AD patients and 3 healthy controls with statistical comparisons using t-test (mean) and F-test (variance). **b**, Violin plots of SH and TEWL measurements from individual participants (AD1–3, Normal1–3), highlighting inter-individual variability within each group. Correlation between SH and TEWL for **c**, atopic dermatitis patients and **d**, normal subjects.

Comment (9) Long-term monitoring is indeed not friendly to the diseased skin of AD patients. So, I suggest improving the test method to prove its effectiveness for skin disease detection, such as using the analyzer to perform a single, short-term measurement on the lesional skin.

Our response:

We would like to express our appreciation to the reviewer for providing valuable feedback. As noted in the reviewer's comment, directly measuring lesional skin in AD patients may be harmful to skin health. Therefore, in this study, we conducted measurements on non-lesional sites adjacent to the lesions. Previous studies [S. Seidenari et al., *Acta dermatovenereologica*, 75. 429, 1995], [B. Eberlein-König et al., *Acta dermatovenereologica*, 80, 188, 2000], [F. Addor et al., *International journal of dermatology*, 51, 672, 2012] demonstrated that evaluating skin parameters near lesional areas is sufficient for diagnosing atopic dermatitis.

Moreover, Figure L5 [Le Fur, I. *et al.*, 11, 193 ,2001] demonstrated that skin exhibits circadian behavior, which is a critical factor to consider. Our findings show that key skin indicators such as TEWL and skin hydration (SH) also display circadian patterns. Thus, long-term monitoring provides more clinically meaningful insights than short-term measurements.

Nevertheless, as the reviewer pointed out, minimizing contact with the skin is essential for maintaining skin health in AD patients. Although various non-invasive methods such as diffuse reflectance spectroscopy (DRS) [Ying-Yu Chen, et al. Non-invasive assessment of skin hydration and sensation with diffuse reflectance spectroscopy. *Scientific reports*, 2023] and nuclear magnetic resonance (NMR) [Ella R. Shilliday, et al. Single-Sided Nuclear Magnetic Resonance (NMR) for the Analysis of Skin Thickness and Collagen Structure in Scarred and Healthy Skin. *Applied Magnetic Resonance*, 2023] techniques have been developed for measuring skin health, their integration into wearable devices remains limited due to high battery consumption and the bulky size of the measurement systems. Therefore, there is a need to develop non-invasive, non-contact wearable systems capable of real-time monitoring skin health without compromising the skin barrier. We have added this direction as future research direction in the Discussion section.

Figure L5. Previous study of circadian variations of TEWL on the cheeks. TEWL was measured on the left cheek of the eight study subjects at 4 h intervals during 48 h. As no variations of TEWL measurements were found between the 2 d, data were pooled on a 24 h basis. For each subject, time point values were expressed as percentages of 24 h individual mean. Then, the mean values of these variations for the study sample (*black squares*, mean \pm SEM) were displayed to express time-dependent changes of TEWL on the cheek (plexogram). Time dependence was detected with two peaks at 8:00 and 16:00 and a trough between 20:00 and 0:00. Analysis by the cosinor method detected a circadian rhythm and provided the best-fitting curve that models the circadian variations for the 24 h period. This curve (*dotted line*) is superimposed on the corresponding plexogram. The light off period is indicated as a bold line on the time axis.

Our modification to the manuscript:

(Page 21 line 485-491)

There are still opportunities for the following improvements to our long-term monitoring device: i) Miniaturization of the device targeted for infancy and children, ii) Integration of additional sensors to measure external environmental factors that can affect the skin (e.g., particulate matter, nitrogen dioxide, formaldehyde), iii) Further clinical studies to

quantitatively analyze factors affecting the skin over the long term, **iv) Integrating wearable devices with non-invasive measurement methods (e.g., diffuse reflectance spectroscopy) to minimize damage to patient skin.**

Reviewer: 2

This work propose a wearable device with size comparable to that of a smartwatch. The device aims to measure the skin humidity (SH) and transdermal water loss (TEWL). This is achieved by a micromechanical system (MEMS) that includes thermistors and heater for SH, a breathable chamber and a humidity sensor for TEWL. The work includes a 28-day validation on three patients. The complexity and the originality of the device's design and the clinical validation may be worth of publication. The reviewer has, however, some comments which may clarify some points and improve the paper. Briefly:

Our response:

We sincerely thank the reviewer for the thoughtful comments and evaluation of our work. The suggestions raised valuable points regarding the device's sensing mechanisms, reliability, and technical details. Accordingly, we have expanded the main text to describe the working principles of both the skin-hydration and TEWL sensors, clarified the role of the SMA-driven bistable actuator with new figure panels, and added a component-level power-consumption analysis. We additionally specify all sensor and electronic part numbers and report mechanical-simulation results that confirm the device's robustness under expected on-skin loads. We have carefully addressed each comment and revised the manuscript to provide clearer explanations and additional data as requested.

Comment (1) The manuscript is well written however additional details on the working principle of the sensors should be included in the main text (part of note 1 and 2 for example);

Our response:

We sincerely appreciate your constructive input and insightful comments on our manuscript. To improve reader understanding, we included the main text by including key content from the Supplementary Note 1 regarding the sensing mechanism. The principle for measuring skin hydration follows a sequence of steps. First, temperatures at the heater location and at a point 1.5 mm away are measured using a Wheatstone bridge. Based on the resulting temperature difference, the thermal properties of the skin are calculated using the finite element method (FEM). The skin hydration is then estimated from these thermal properties using the Maxwell–Eucken model. Additionally, to allow calibration for each fabricated sensor, we used S184 and S170 polymers. Interpolation was performed based on the known thermal properties and corresponding hydration levels of these materials.

We thank the reviewer for the valuable comment. In response, we have added a detailed explanation of the working principle and calibration process of the TEWL sensor to the main text (Page 7, Line 140–148). Specifically, we describe how TEWL is calculated by analyzing the slope of humidity increase (k) within the closed chamber, and how this slope is calibrated against reference values obtained using the standard wet-cup method. Reference TEWL values were measured at water temperatures of 23 °C and 40 °C, and the corresponding slope values were used to derive the calibration factor (CF). We believe these additions clarify the sensing mechanism and allow readers to understand the core concept without relying solely on the Supplementary Note.

Our modification to the manuscript:

(Page 7 line 140-148)

The SH sensor includes two pairs of NTC temperature sensors and a pair of heaters. One pair of NTCs (NTC 2) is placed above the heaters, while the other (NTC 1) is located 1.5 mm away to form a temperature gradient reference (Supplementary Fig. 4). When the SH sensor contacts the skin, the SH sensor applies heat to the resistance heating element (heater) and measures the

temperature (T_1 :NTC 1, T_2 : NTC 2) at NTCs. The difference between the two temperatures ($\Delta T = T_2 - T_1$) helps eliminate ambient temperature fluctuations, ensuring reliable moisture assessments (Supplementary Fig. 5). Using transient heat transfer, we derive the thermal characteristics of the skin, such as thermal conductivity (k) and thermal diffusivity (α), and from these, the skin's moisture content (ϕ_s). ~~The thermal response of the skin was measured using Wheatstone bridges and NTC thermistors. Based on the measured thermal response, the thermal properties of the skin were calculated through finite element analysis and the Maxwell-Eucken model, enabling the estimation of skin hydration levels.~~

(Page 8 line 175-186)

We validated the TEWL sensor by analyzing humidity changes across a semi-permeable membrane (Fig.2f), ~~with detailed information provided in Supplementary Note 2.~~ When the chamber is closed by a vertical load, the relative humidity increases to 29 %. After 25 seconds of being closed and upon removal of the external force, the chamber opens and the humidity drops back to the initial humidity value of 23 % through ventilation via the hole of the chamber. TEWL is calculated by multiplying the slope of increasing humidity (k), determined to be 0.42 %/s within the 6 to 9 second interval, where linearity reaches 99.5% as assessed using the least squares (LS) method, with a calibration factor (CF) as shown in Fig. 2g³³. ~~To determine CF, we applied the wet-cup method to measure the k value, which was matched to reference TEWL values obtained using water at 23 °C and 40 °C (5.1 ± 0.4 and 23.7 ± 0.9 g/m²/h, respectively). Detailed procedures are provided in Supplementary Note 2. The corresponding k values were 0.21 and 0.98, from which the calibration factor was calculated using $TEWL = k \times CF$, yielding an approximate value of 24. CF was obtained using the wet-cup method, a representative method for calibrating commercial TEWL sensors.~~

Comment (2) The MEMs operating principle is not totally clear to the reviewer. the role of SMA should be better explain. why SMA wires are needed? what is the physical mechanism behind the operation?

Our response:

We agree that operating principle and role of SMA actuator should be explained in more detail for better understanding. In the Introduction, we have further explained why SMA wire is needed in this compact device in terms of its high power density. Additionally, we have included new figure panels (Figure 2d, 2e and 2f) to present side view and top view of the actuator, as well as the energy-displacement profile of the bistable beam for a more detailed explanation of the physical mechanism.

Our modification to the manuscript:

(Intro...)

For continuous and accurate measurement and ventilation cycles, the sensor module and the surrounding chamber require a compact actuator capable of repeatedly attaching them to and detaching them from the skin. To ensure compactness and energy efficiency for long-term use, we implemented a shape memory alloy (SMA) wire with a bistable beam. SMA, with its exceptionally high power density of up to 5 kW/kg, is a promising candidate for integration into compact wearable device.

(line 223..)

SMA wires are placed either on the upper or lower side of the compliant beam with the interlocking joints (Fig. 3d and 3e). Each actuator triggers snap-through bistable transition between the beam's downward and upward stable states (Fig.3f).

Figure 3. Mechanical behavior analysis between the breathable chamber and the skin, and the corresponding actuator design. **c**, Image of the actuator combining a bistable beam, SMA wire, chamber, and frame. **d**, Side view, and **e**, top view of the actuator in its initial state before being fit into the frame. Upper and lower SMA wires are mounted on both sides of the beam. **f**, Energy-displacement curve of the bistable beam, showing two stable minimum energy state (① and ②) and an intermediate unstable state (snap-through). **g** Schematic of the side view of the actuating part demonstrating the actuation and sensing mechanism.

Comment (3) Fig1.b shows a magnet and a cap. The reviewer misses the explanation of their functions.

Our response:

We thank the reviewer for pointing out the missing explanation regarding the magnet and cap in Fig. 1b. The cap serves as a mechanical enclosure that integrates the sensor FPCB and the breathable chamber, forming a unified sensor part (Supplementary Fig. 3). Two magnets embedded in the cap interface with corresponding magnets in the actuator, enabling magnetic self-alignment and secure attachment. This magnetic coupling not only ensures precise positioning between the sensing module and actuator, but also allows for tool-free assembly and easy replacement without compromising alignment or structural integrity. We have revised the manuscript accordingly to include this explanation.

Supplementary Fig. 3 | Fabrication method of the BSA sensor.

Our modification to the manuscript:

(Page 5 line 93-102)

The BSA comprises a sensor module, control board, actuator, all compactly and comfortably worn on the body using a low-irritation strap band compatible with commercial (Fig. 1b and c). For easy maintenance, the board and sensor module are connected via a standard board to board connector, and the sensor module, comprising a chamber and sensor FPCB integrated within the protective cap, is physically attached to and aligned with the actuator using two pairs of magnets, allowing rapid replacement without damage. To achieve compactness and energy efficiency for long-term use, we implemented a shape memory alloy (SMA) wire with a bistable beam.

Comment (4) The complexity of the micromechanical system highlights the ability of the involved research groups to engineer and craft wearable devices. However, complexity also raises concerns about reliability especially for wearable-on skin MEMS devices where sweat, movements (friction etc..) may bring to failure. Can the authors comment on this? Did they run any simulation to evaluate the stress arising during operation (snapping and release)?

Our response:

We thank you for your comment on the reliability of our device. For clarity, we classify possible mechanical failures into two: (i) external stimulus to the sensor module while contacting the skin, and (ii) high stress in SMA wire during operation.

First, two external stimuli—sweat and excessive compressive load—could threaten the sensor module. The sensor module is encapsulated in a 408 μm elastomer layer (Fig. R1a), which blocks sweat preventing damage to sensor components such as the heater, NTC and copper traces. Regarding an external force, the rigid plastic cover primarily protects actuator and sensor components from all the direct external forces (Fig.1b). Still, if the device is pressed hard against the skin while the sensor is in its measurement state, the reaction force acts directly on the sensor. This may not affect the sensor when the sensor is not in contact with the skin (ventilation state). Only the rigid aluminum frame is in contact with the skin in this state.

To set an upper limit on the load the sensor can experience, we consider the critical force of the bistable structure that is connected to the sensor. As external pressure increases, the reaction force applied to the sensor rises until it reaches this critical force, at which point the beam snaps to its alternate stable state. Thus, the sensor is only allowed to experience more than the beam's critical force. In the experiment, the maximum critical force is measured 1.2 N.

We simulated the sensor module under a 1.2 N compressive load (Supplementary Fig. 12a), with heater and NTC elements covered by the elastomer layer and in contact with a skin model. The simulation results show that the relatively stiff sensor component (heater and NTC) has negligible stress and strain under 1.2 N of pressing (Supplementary Fig. 12b and c). Soft elastomer layer covering the sensor components shows maximum local strain of $\sim 7\%$, far below its maximum strain ($\sim 900\%$).

To summarize, the bistable beam limits the reaction force on the sensor at 1.2 N, and under this load all the sensor components remain well within their mechanical limits. Hence, the sensor module is robust against both sweat ingress and excessive pressing forces.

Second, SMA wire used in our device experiences its high stress during the snap-through of the bistable beam. Since the wire is as thin as the human hair, it is important to select an appropriate wire diameter and placement on the beam to keep the peak stress below the material's yield strength. We quantified the stress for different wire configurations by measuring the force required for the snap-through (Fig. 3k and Supplementary fig.11). By varying the wire diameter and the number of wire arrays of the wire we distributed the load and identified an optimal configuration: two parallel wires, each 0.1 mm in diameter. Cyclic testing of this configuration confirmed that the wire withstands more than 2000 actuation cycles under the designed operation conditions (Fig.3m).

Supplementary Figure 12 | Finite-element simulation of the sensor module on skin under normal **a**, Schematic illustration of the sensor components and skin layers. **b**, Stress distribution in the sensor module subjected to a 12 N normal pressing load. **c**, Strain distribution under a 12 N normal pressing load.

Figure 3. k, Stress profile as a function of distance based on the characteristics of the SMA wire. m, Average contact force of the bistable beam over cycles.

Supplementary Figure 11 | Illustration of the Free body diagram of SMA wire.

Comment (5) The authors do not comment on the power consumption of the device. This would be something interesting for the readers to know.

Our response:

We sincerely appreciate your constructive input and insightful comments on our manuscript. In response to the reviewer's comments regarding power consumption, we have enhanced the reproducibility and overall quality of our sensor system. To enable long-term monitoring, we ensured a minimum operation time of 24 hours, supported by both theoretical estimation and experimental validation. All components used in our device are listed in Supplementary Table 5, as referenced in Comment 6.

The selected battery has a rated voltage of 3.7 V, a capacity of 370 mAh, and an energy of 0.925 Wh. The main power consuming components are the heater, actuator, Bluetooth module, and sensor, while components such as capacitors and inductors were assumed to have negligible power consumption. Power consumption was estimated based on the dominant contributors: the heater and actuator were calculated using Ohm's law ($I = V/R$), whereas the Bluetooth module and sensor were assessed based on their datasheets. The heater and actuator operate intermittently (10 seconds and 1 seconds per 5 minutes, respectively), while the Bluetooth module and sensor are assumed to operate continuously. The estimated power consumption of each component by an hour is as follows.

Heater (2ea, 12times actuation = 120s = 0.033h, 100 Ω , 3.7V)

$$: P = \frac{V^2}{R} = \frac{3.7V^2}{100\Omega} = 0.1369W, (power\ consumption) = 9.13mWh$$

Actuator (12times actuation = 12s = 0.0033h, Shape Memory Alloy 7.15 Ω , 2.7V)

$$: P = \frac{V^2}{R} = \frac{2.7V^2}{7.15\Omega} = 1.0196W, (power\ consumption) = 3.36mWh$$

Bluetooth module (datasheet, duty cycle 5%)

$$: P = 1.85mW, (power\ consumption) = 1.85mWh$$

SH Sensor (4ea, 10k Ω)

$$: P = \frac{V^2}{R} = \frac{3.7V^2}{10k\Omega} = 1.369mW, (power\ consumption) = 5.48mWh$$

TEWL Sensor (datasheet)

$$: P = 2.96mW, (power\ consumption) = 2.96mWh$$

The total power consumption was estimated to be 22.78 mWh per hour, resulting in approximately 546.72 mWh over 24 hours. Based on this, the system was designed with a safety factor of approximately 1.7.

In actual operation, the BSA sensor's battery voltage was measured to decrease from 4.05 V (fully charged) to 3.67 V after 24 hours of use. This corresponds to a remaining battery capacity of approximately 55.83%, calculated as $\frac{3.67-3.0}{4.2-3.0} = 55.83\%$. Therefore, both theoretical estimation and experimental results confirm that the device operates reliably for at least 24 hours.

Power consumption

Component	Actuation time (per hour)	Power (W)	Energy consumption (Wh)
Heater	0.033	$P = \frac{V^2}{R} = \frac{3.7V^2}{100\Omega} = 0.1369W$	9.13mWh
Actuator	0.0033	$P = \frac{V^2}{R} = \frac{2.7V^2}{7.15\Omega} = 1.0196W$	3.36mWh
Bluetooth module	0.05	$P = 1.85mW$	1.85mWh
SH sensor	1	$P = \frac{V^2}{R} = \frac{3.7V}{10k\Omega} = 1.369mW$	5.48mWh
TEWL sensor	1	$P = 2.96mW$	2.96mWh

(Page 13 line 300-305)

The contraction of the SMA 1 for 0.5seconds results in the downward actuating of the bistable beam, which closes the breathable chamber. TEWL is then measured over a 10 second period. Subsequently, the heater is activated for an additional 10 seconds to measure SH. After completing these measurements, the voltage is applied to trigger SMA 2 for 0.5 seconds which switch the bistable structure to bend upward.

Comment (6) Methods section should include the exact models of the chip used (NTC, heaters, etc.). is the humidity sensors a commercial one?

Our response:

We thank the reviewer for the valuable comment. The humidity sensor used in the TEWL sensor is a commercial sensor. In response, we have clearly added descriptions and specific manufacturer part numbers of the sensor components in both the Results and Methods sections. Detailed information on the sensor components is provided in the top section of Supplementary Table 5, while the list of electronic components related to wireless communication and the battery management system is presented in the bottom section of the same table. These revisions are expected to enhance the clarity of the device configuration and improve the reader's understanding of the BSA board design.

Our modification to the manuscript:

(Page 7 line 140-145)

The SH sensor includes two pairs of NTC temperature sensors and a pair of heaters. One pair of NTCs (NTC 2) is placed above the heaters, while the other (NTC 1) is located 1.5 mm away to form a temperature gradient reference (Supplementary Fig. 4). When the SH sensor contacts the skin, the SH sensor applies heat to the resistance heating element (heater) and measures the temperature (T_1 :NTC 1, T_2 : NTC 2) at NTCs. **Details of the sensor components are provided in the top section of Supplementary Table 5.**

(Page 13 line 290-297)

Figure 4 shows the wireless system and demonstrates the verification of the BSA performance through an in vivo test. The BSA consists of an FPCB (**details of the board design in Supplementary Fig. 13 and the bill of materials for all electronic components is provided in the bottom section of Supplementary Table 5**) that includes a Bluetooth chip with an integrated SH

sensor, TEWL sensor, a cover that protects the FPCB for everyday use and actuator, powered by a 250 mAh lithium-ion battery (maximum operating time of 36 hours), as shown in Fig. 4a. The robust design and integrated system minimize user fatigue during long-term monitoring, allowing convenient measurements without complex protocols or specialized training.

(Page 23 line 556-572)

Sensor fabrication

The sensor is made in nine steps, as shown in Supplemental Fig. 3. The FPCB containing the SH and TWEL sensors is attached to the 3D printed cap (a). **The FPCB includes a pair of resistance heating elements (ERJ-3EKF1000V), two pairs of NTC thermistors (NCP03XH103J05RL) and a commercial humidity sensor (SHT40-AD1B-R2).** The polymer layer that contacts the skin is made in two steps. The first layer is polymer mix of Eco-flex 0030 and Eco-flex gel in a 1:1 ratio, which is poured onto the acrylic and spin coated at 800 rpm for 30 seconds (b). The evenly coated polymer layer is semi-cured for 4 minutes at 70°C (c). The second layer is for attaching the polymer to FPCB. Silicon adhesive is poured over the first layer and spin-coated at 500 rpm for 30 seconds (d). The evenly coated silicon adhesive is semi-cured for 4 minutes at 120°C (e). Silicon adhesive is also coated in the same way and semi-cured. Adjust the height of the acrylic structure so that only the FPCB with the sensor fabricated in (a) is attached to the second layer and allow it to cure completely (f). A mold is made on a 3D printer to fit the shape of the chamber, which consists of an upper and lower part. Dragon skin 0030 is poured into the mold and fully cured to produce the chamber (g). The sensor part made in step (f) is attached to the chamber (h). Finally, attach the two sensor parts, one with the N pole upward and the other with the S pole upward (i).

a. Skin hydration & TEWL sensor part

Component	Quantity per sensor	Description	Manufacturer part number
NTC1+/ NTC-1	2 / 2	THERMISTOR NTC 10KOHM 3380K 0201	NCP03XH103J05RL
Heater	2	RES SMD 100 OHM 1% 1/10W 0603	ERJ-3EKF1000V
SH1	1	SENSOR HUMIDITY 100 RH SMD	SHT40-AD1B-R2
CN1	2	CONN PLUG 10POS SMD GOLD	505274-1012

b. Wireless communication & battery management system

Component	Quantity per sensor	Description	Manufacturer part number
C1	2	CAP CER 12PF 50V C0G/NP0 0402	GJM1555C1H120FB01D
C2	3	CAP CER 10UF 6.3V X5R 0603	CL10A106KQ8NNNC
C3	1	CAP CER 1UF 50V X7R 0805	CL21B105KBFNNNE
C4	4	CAP CER 2.2UF 16V X5R 0603	CL10A225KO8NNNC
C5	2	CAP CER 0.1UF 25V X7R 0402	CL05B104KA5NNNC
C6	1	CAP CER 1UF 10V X7R 0603	CL10B105KP8NNNC
C7	3	CAP CER 1UF 6.3V X7R 0402	CL05B105KQ5NQNC
D1	1	LED BLUE CLEAR 0603 SMD	150060BS75000
L1	1	FIXED IND 10UH 300MA 600MOHM SMD	MLZ1608N100LT000
L2	1	FIXED IND 15NH 460MA 0.16OHM SMD	LQW15AN15NJ00D
R1	9	RES 10K OHM 1% 1/10W 0603	RK73H1JTTD1002F
R2	1	RES 1K OHM 1% 1/10W 0603	RMCF0603FT1K00
R3	1	RES 6.04K OHM 1% 1/10W 0603	RMCF0603FT6K04
R4	1	RES SMD 1.35KOHM 0.1% 1/10W 0603	RT0603BRD071K35L
R5	1	RES 0 OHM JUMPER 1/10W 0603	RC0603JR-070RL
R6	2	RES 4.22K OHM 1% 1/10W 0603	RMCF0603FT4K22
SW1	1	SWITCH SLIDE SPDT 25MA 24V	EG1215AA
U1	1	RF TXRX MOD BLUETOOTH CHIP SMD	MDBT42V-512KV2
U2	1	IC REG LINEAR 3.3V 150MA 6-WSON	TPS70933QDRVRQ1
U3	1	IC BATT CHG LI-ION 1CELL 6DSBGA	BQ25100YFPR

U4	1	CONN RCPT USB2.0 MICRO B SMD R/A	0473460001
U5	2	MOSFET N-CH 12V 2.9A 3PICOSTAR	CSD13383F4T
U6	1	IC REG LINEAR 1V 300MA 6-WSON	TPS7A1010PDSET
X1	1	CRYSTAL 32.7680KHZ 12.5PF SMD	ECS-.327-12.5-34B-TR
CN1	1	CONN RCPT 10POS SMD GOLD	5052701012

Supplementary Table 5 | Bill of Materials. a, Skin hydration & TEWL sensor part. **b,** Wireless communication & battery management system.

Reviewer: 1

Comment (1) The authors have satisfactorily addressed all my concerns. However, I think a recent work on skin health monitoring published in Nature Communications (An Epidermal Serine Sensing System for Skin Healthcare, Nature Communications, 2025, 16, 2681.) should be cited and If the author wants to claim the system's key advantage, they should analyzed for its differences from this work.

Our response:

We sincerely appreciate the reviewer's constructive inputs and insightful comments on our manuscript. The referenced study focuses on a device that measures the concentration of serine in the skin, a biomarker that can be used to evaluate skin barrier function. In our study, we developed a system designed to measure additional key indicators of skin barrier integrity, such as trans epidermal water loss (TEWL) and skin hydration (SH), without requiring active user operation. Compared to the referenced work, our system offers advantages in terms of long-term monitoring capability, skin ventilation, and wearable integration. Accordingly, in the Discussion section, we proposed a future research direction to integrate additional skin-contact-based sensors into the device for monitoring other epidermal biomarkers, such as serine concentration and sweat pH.

Our modification to the manuscript:**(Page 21 line 494-495)**

There are still opportunities for the following improvements to our long-term monitoring device: i) Miniaturization of the device targeted for infancy and children, ii) Integration of additional sensors to measure external environmental factors that can affect the skin (e.g., particulate matter, nitrogen dioxide, formaldehyde), iii) Further clinical studies to quantitatively analyze factors affecting the skin over the long term, iv) Integrating wearable devices with non-invasive measurement methods (e.g., diffuse reflectance spectroscopy) to minimize damage to patient skin, v) **Development of a device for measuring additional skin**

health biomarkers (e.g., epidermal serine concentration ⁵⁷, sweat pH ⁵⁸). We envision that this long-term skin health monitoring system will enable individuals with sensitive skin to provide moisture or block external environmental factors based on quantitative indicators, thereby improving skin health care. Moreover, it could be used to identify factors that have long-term adverse effects on the skin barrier, which current technology has yet to elucidate.

(Reference)

- 57 Yuan, Y. *et al.* An epidermal serine sensing system for skin healthcare. *Nature Communications* **16**, 2681 (2025).
- 58 Nakata, S. *et al.* A wearable pH sensor with high sensitivity based on a flexible charge-coupled device. *Nature Electronics* **1**, 596-603 (2018).